# A latitudinal gradient of deep-sea invasions for marine fishes

Sarah T. Friedman [1,2] ✉ & Martha M. Muñoz[1]

Although the tropics harbor the greatest species richness globally, recent work has demonstrated that, for many taxa, speciation rates are faster at higher latitudes. Here, we explore lability in oceanic depth as a potential mechanism for this pattern in the most biodiverse vertebrates – fishes. We demonstrate that clades with the highest speciation rates also diversify more rapidly along the depth gradient, drawing a fundamental link between evolutionary and ecological processes on a global scale. Crucially, these same clades also inhabit higher latitudes, creating a prevailing latitudinal gradient of deep-sea invasions concentrated in poleward regions. We interpret these findings in the light of classic ecological theory, unifying the latitudinal variation of oceanic features and the physiological tolerances of the species living there. This work advances the understanding of how niche lability sculpts global patterns of species distributions and underscores the vulnerability of polar ecosystems to changing environmental conditions.

Understanding patterns of biodiversity across large-scale gradients, and the processes that underpin them, is a primary goal of evolutionary biology. One of the most conspicuous spatial trends is the latitudinal diversity gradient, wherein species richness increases towards the tropics[1,2]. This pattern is common in terrestrial and marine ecosystems, through geological time, and across many phylogenetic groups[1,3,4]. Faster speciation in the tropics is often invoked to explain this conspicuous diversity gradient, and studies on amphibians[5], plankton[6], plants[7], and mammals[8] all support this notion. Yet, more recent work has demonstrated that speciation rates may not peak at low latitudes and can even exhibit an inverse latitudinal gradient in some taxa, leading to a spatial mismatch between species richness and the rate at which such diversity arises[9–12]. Given that latitudinal patterns of speciation rates are not consistent across lineages, the processes that generate rapid speciation, particularly at high latitudes, remain elusive.

We focus on fishes, the most biodiverse vertebrates, which exhibit a traditional latitudinal diversity gradient defined by greater species richness in tropical regions[9]. Yet, paradoxically, the five marine fish lineages with the fastest speciation rates are not found in the tropics, but rather at high latitudes, regions characterized by high endemism[9]. Snailfishes, rockfishes, flatfishes, and eelpouts have all proliferated extensively in northern poleward regions, while icefishes and their allies dominate Antarctic waters. With the same high-latitude regions also experiencing the fastest warming rates[13], the effects of human-induced global warming are poised to rapidly and catastrophically affect these ecosystems[14]. Thus, the lineages that are most quickly accumulating marine fish biodiversity are also those likely to be most imperiled by global climate change[14]. Discovering the mechanisms that drive rapid diversification in these lineages is key to both understanding how species arise and how to maintain biodiversity in the face of a changing environment. Here, we explore diversification across the depth gradient as a potential factor that may have allowed fish lineages to rapidly dominate high-latitude marine communities.

Understanding how depth lability might contribute to speciation rate requires exploring the variables associated with the deep sea, and how they vary across latitude. Marine fishes occupy an incredible range of depths, from the near-shore intertidal zone to some of the deepest oceanic regions at nearly 8000 m[15]. Home to some of the most fantastical creatures, the deep sea is characterized by crushing pressures, a complete absence of ambient light, and extremely cold temperatures. Inevitably, these environmental parameters vary substantially along the depth gradient from shallow seas to the abyssal plains[16,17]. Reflecting the environmental gradient and the associated physiological challenges across these depths are broad shifts in the vertical

[1]Department of Ecology and Evolutionary Biology, Yale University, New Haven, CT 06511, USA. [2]Yale Institute for Biospheric Studies, Yale University, New Haven, CT 06511, USA. ✉e-mail: sarahtfried@gmail.com

structure and evolutionary dynamics of marine communities[18–22]. Rates of morphological evolution accelerate in deep-sea fishes, likely reflecting the relaxation of locomotion performance pressures[23,24]. These findings suggest that shifts in the environmental features along the depth gradient can have significant consequences for fish evolution and diversification. Yet, there is also latitudinal variation in the vertical distribution of environmental parameters: temperature and density are less stratified at high latitudes[25], which may relax abiotic barriers to diversification along the depth gradient. Studies that examine the patterns and processes driving latitudinal fish diversity have largely neglected the depth axis[26,27], reflecting a historic lack of data, particularly for deep-sea fish species. We argue that environmental interactions structured across depth and latitude may represent a critical, yet understudied, axis of fish diversification.

Although studies have indicated that speciation rates are not faster in deep-sea fishes[9], none so far have examined the direct interplay between latitudinal variation, speciation rate, and ecological lability along the depth gradient. Clades that can freely disperse along the depth axis may be more likely to capitalize on novel resources or niches along that gradient and become isolated by (vertical) distance, leading to repeated local adaptation and speciation events. By contrast, a lineage with relatively limited depth lability would not be expected to experience the same increased speciation rate. In essence, speciation rates would be positively correlated with rates of depth evolution. Although studies have indicated that niche lability can drive high diversification rates in birds[28–30], lungless salamanders[31], and irises[32], none have examined if these same principles apply to the depth gradient of marine organisms. Moreover, niche lability could alter community composition in a predictable way such that shallow and deep communities of fishes may not substantially differ (phylogenetically) in regions that are dominated by a few clades that also exhibit elevated rates of depth evolution. Niche lability may correspondingly shape broader ecosystem dynamics and species interactions by latitude.

Here, we explore the relationship between speciation rates, depth lability, and latitude, with a particular emphasis on five rapidly speciating lineages: icefishes and their allies (Notothenioids), snailfishes (Liparidae), rockfishes (Sebastidae), flatfishes (Pleuronectidae), and eelpouts (Zoarcidae)[9]. These lineages exhibit elevated speciation rates among ray-finned fishes, and species within these clades inhabit a wide range of habitats from tidepools to the depths of oceanic trenches. We hypothesize that (1) the rapid speciation rates observed in these five lineages are driven by elevated rates of depth evolution and (2) species at higher latitudes exhibit greater depth lability facilitated by the reduced stratification of environmental variables in polar regions. We then evaluate the ecological and evolutionary consequences of latitudinal bias in depth lability across ray-finned fishes on a global scale. We hypothesize (3) that the latitudinal patterning of deep-sea invasions left a fingerprint on global community assembly by homogenizing phylogenetic community structure throughout the water column at higher latitudes. Our findings indicate that diversification along the depth gradient has played a major role in the success of high-latitude fishes, cementing niche lability as a fundamental feature of global fish distribution and diversity.

## Results

### Depth lability and speciation in focal clades

In high-latitude, rapidly speciating clades (Liparidae, Pleuronectidae, Sebastidae, Zoarcidae, and Notothenioids) transitions along the depth gradient exceeded those expected based on their age and species richness (Fig. 1). This pattern extended to several other families, such as lanternfishes (Myctophidae), jacks (Carangidae), and groupers (Serranidae), which also exhibited extreme lability along the depth gradient, implying that rampant depth transitions are not exclusive to the fastest speciating lineages. Most other families, however, do not transition along the depth gradient any more than expected given their

age and species richness. Notable among these are hyper-diverse lineages that dominate coral reefs: wrasses (Labridae), damselfishes (Pomacentridae), and gobies (Gobiidae), for which transitions to deeper waters are exceptionally rare. Using a linear regression, we find significant support for more transitions along the depth gradient in higher latitude clades ($F = 8.83$, $p = 0.014$), though, as described above, speciation rates are not higher in all clades that exhibit depth lability (Fig. 1). These findings exhibit phylogenetic structure, as a PGLS between the same factors was not statistically significant ($p > 0.05$), meaning that depth transitions are biased to a related subset of total fish clades. We find significant support for a relationship between speciation rate and the rate of depth transitions using a PGLS approach ($F = 15.09$, $p = 0.001$). These results are consistently significant with a non-phylogenetic regression across all methods of estimating speciation rates (Fig. 2A). Overall, clades primarily comprised of shallow-water species are more likely to exhibit transition rates within or below the expectation. Likewise, with the exception of dragonfishes (Stomiidae), a circumglobal clade, deeper species tend to belong to high-latitude clades that readily transition along the depth gradient. These findings are all robust to re-categorization of the depth categories (Fig. S1) and variation in divergence time estimates (Figs. S2 and S3). In short, exceptional depth transitions are biased to a few, relatively closely related rapidly speciating, high-latitude fish clades.

### Depth and latitude evolution

For our analysis examining the relationship between depth and latitude more generally across fishes, we find strong support for an all-rates-different transition model ($\Delta AIC = 179$). The majority of transitions occur between temperate and polar latitudes at both intermediate and profound depths (Fig. 3, Table S1). Tropical latitudes appear to import diversity from other latitudes, with the majority of transitions occurring either across depths within the tropics or towards tropical regions from higher latitudes. In other words, globally, the vast majority of transitions across the depth gradient occur in high latitudes and between deeper oceanic zones. These results are relatively consistent when analyzed with the Ghezelayagh et al.[33] phylogeny, though we find comparatively more transitions in temperate regions and fewer in polar latitudes (Fig. S4). This is unsurprising, as this phylogeny does not include many of the high-latitude taxa present in the original analyses. For example, after being trimmed to our dataset, there are just two snailfish species and only 30 taxa total that are categorized as polar. Given the paucity of polar species in this phylogeny, the fact that we still recover vastly more transitions in polar zones than in tropical regions implies that the latitudinal bias in deep-sea invasions is not an artifact of the phylogeny.

Using the MuSSCRat model to estimate rates of depth evolution, we find strong support for separate rates associated with each latitudinal region (posterior probability = 1.0). The rate of depth evolution at polar latitudes is 8.6x faster (on average) when compared with depth evolution at temperate latitudes, even while accounting for background rate variation (Fig. 2B). Fish clades at tropical latitudes have exceptionally low rates of depth evolution, on average >50× slower than that of clades from temperate latitudes. There was an average of 509 transitions between latitudinal regions across the phylogeny, and different priors did not substantially influence posterior distributions (Figs. S5 and S6). From this analysis, we also extracted clade-specific rates of depth evolution for each of the 46 clades. Controlling for phylogeny, we find a positive relationship ($F = 7.49$, $p = 0.023$) between rates of depth evolution and speciation rate across clades (Fig. 2B), meaning that in faster-speciating clades evolution across depth categories is also more rapid.

### Global community structure comparisons

Consistent with our expectations, we find a strong latitudinal gradient in community differences between shallow and deep regions

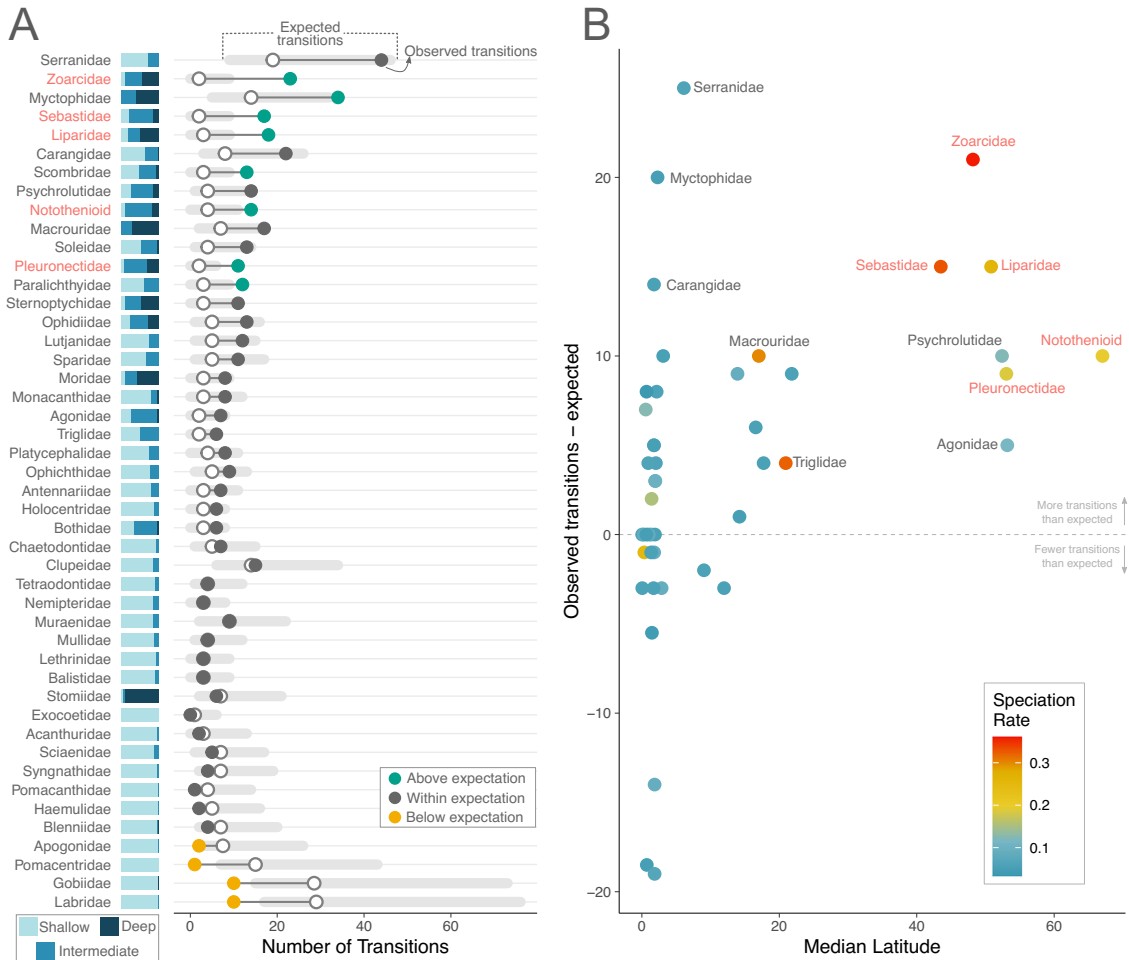

**Fig. 1 | Clade-specific lability along the depth gradient increases with latitude.** The far left (**A**) shows depth composition of extant species in each family. Names in red are the focal high-latitude, rapidly speciating clades. For each clade, the number of transitions between depth categories is compared to a null distribution of depth transitions within that clade (gray region). Open points represent the median null expectation, whereas closed points represent the observed number of depth transitions averaged across 100 stochastic character maps. **B** The relationship between transitions across depth regions (observed transitions minus the median expected number of transitions) and median latitude. Each point depicts a clade and points are colored by family-specific speciation rate, as estimated from Rabosky et al.[9] Source data are provided as a Source Data file.

(Fig. 4 and Fig. S7). Tropical latitudes have significantly greater values of PCD than higher latitudes. Therefore, in tropical latitudes the shallow fish community strongly differs from the deeper fish community, reflecting limited diversification along the depth axis. High-latitude communities appear more phylogenetically homogenous along the depth axis, reflecting repeated invasions of new depth zones. These trends are not driven solely by global patterns in species richness, as we find a relatively weak relationship between species richness and PCD ($r^2 = 0.28$; Fig. S8). We find that PCD values tend to be higher in coastal regions. This pattern may reflect sampling bias, as these regions are more likely to be subjected to fishing efforts, which may inflate the estimates of diversity found in coastal, shallower cells. Continental shelves are also known to coincide with high species diversity[19], particularly in the tropics, which could drive the higher outlier PCD estimates. The contrast between the species diversity of continental shelves and open-ocean regions in combination with the reduced abiotic gradient towards the poles (which we discuss in detail later) is also likely driving the greater variance of PCD estimates found at higher latitudes. Furthermore, we observe a notable decrease in the variance of PCD estimates around −60°, which coincides with a reduction in land mass and continental shelf habitat at this latitude.

## Discussion

The latitudinal diversity gradient, in which species richness is biased towards the tropics, can be matched by faster speciation rates[6,34]. Yet, in marine fishes—the most speciose vertebrate group—species richness is highest in the tropics, whereas speciation rates counterintuitively increase towards the poles[2,9,35]. Here, we examine whether niche lability across depth underpins this widespread pattern in speciation rates. We provide evidence that the depth gradient has been an important driver of high-latitude fish diversification. High latitudes are associated with faster rates of depth evolution across marine fishes, such that the majority of transitions along the depth gradient occur at temperate and polar latitudes and diversity at these latitudes is, in turn, exported to tropical regions (mirroring patterns also found in marine invertebrates[36]). Combined, these trends have shaped latitudinal patterns of phylogenetic community composition along the depth gradient. Our findings reveal that diversification along the depth axis has been a key catalyst for the success of high-latitude fishes and underscore the importance of niche lability in spurring diversification.

### Depth lability and speciation
Supporting our hypothesis, enhanced niche lability across depth is associated with accelerated speciation rates. Rapidly speciating fish

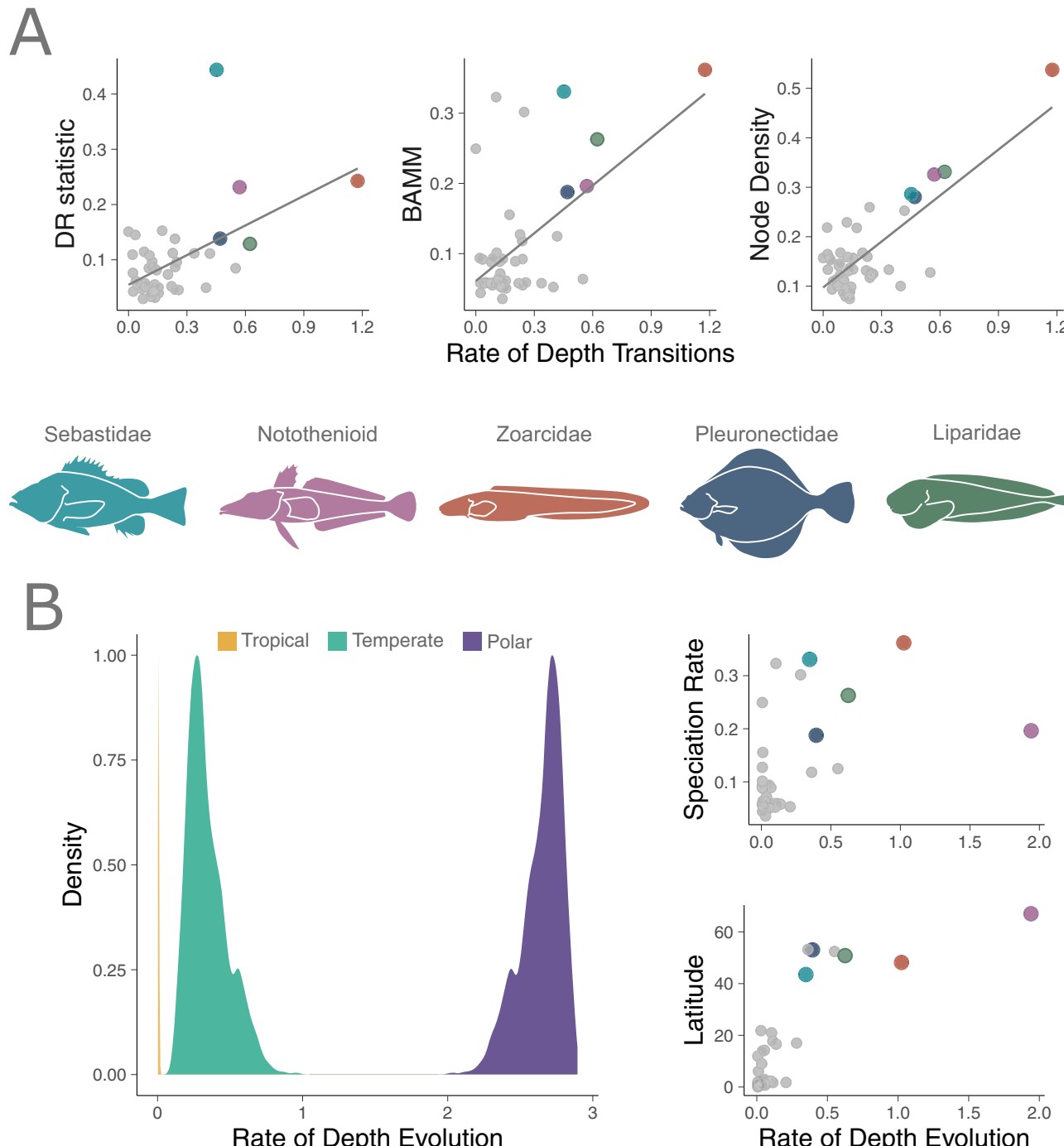

**Fig. 2 | Positive relationship between latitude, rates of depth evolution, and speciation rate largely fueled by the five focal lineages.** The relationship between different speciation rate metrics and the clade-specific rate of depth transitions (**A**). Focal clades are color coded on each plot. Results from the state-dependent relaxed-clock model of Brownian motion showing the density distributions of rates by latitude and the clade-wise relationship between MuSS-CRat depth rates, speciation (BAMM), and latitude by clade (**B**). There is substantial support for independent rates of evolution associated with different latitudes. Source data are provided as a Source Data file.

clades[9]−eelpouts (Zoarcidae), rockfishes (Sebastidae), flatfishes (Pleuronectidae), icefishes and their allies (Notothenioids), and snailfishes (Liparidae)−readily diversify along the depth axis, with more transitions across depth zones than expected based on clade age and species richness. Correspondingly, lability along the depth axis may trigger rapid speciation, a result biased to clades inhabiting higher latitudes[9]. Similar relationships between speciation rate and niche lability have also been reported in several terrestrial lineages, including plants[32], salamanders[31], tanagers[29], and more

broadly across birds[28]. These findings are consistent with the view that rapid adaptation to novel environmental regimes drives diversification. Diversification along any abiotic or biotic axis at different depths may drive specialization, local adaptation, and niche differentiation, in turn catalyzing the speciation process[37−39]. Niche evolution can also facilitate range expansion (vertically, in this case), which can increase the likelihood of vicariance events[40]. We emphasize that these factors are not mutually exclusive and, particularly given the broad scale of this study, it is very likely that

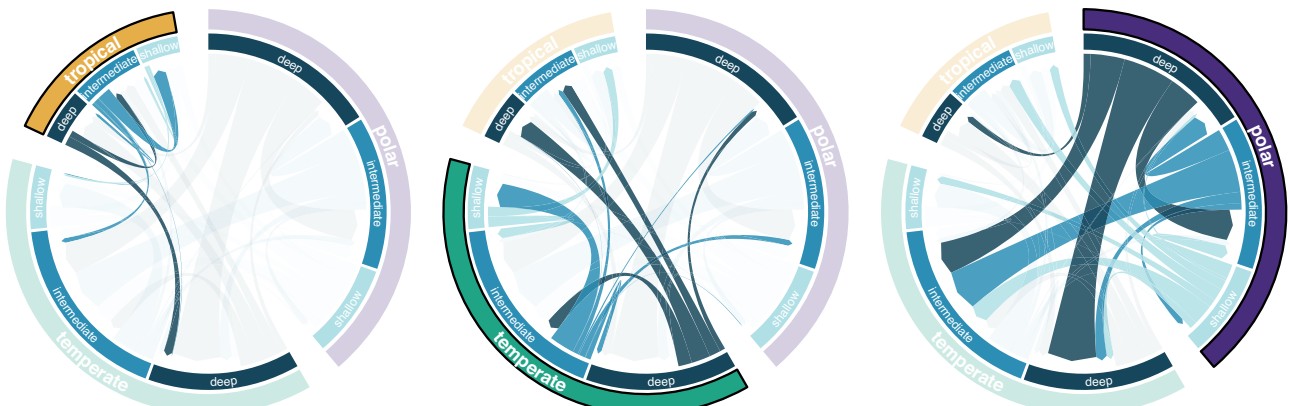

**Fig. 3 | Invasions of the deep sea are predominantly found in high latitudes, concordant with a greater number of transitions across the depth gradient.** Directionality of transitions across both depth and latitude estimated from the best-fit Mk model (ARD). Maximum chord width corresponds to the estimated transition rate (q) and arrow color designates the depth category. Each panel highlights transitions from a different latitude. Globally, the majority of transitions along the depth gradient are found in polar regions, while the tropics exhibit relatively few depth transitions.

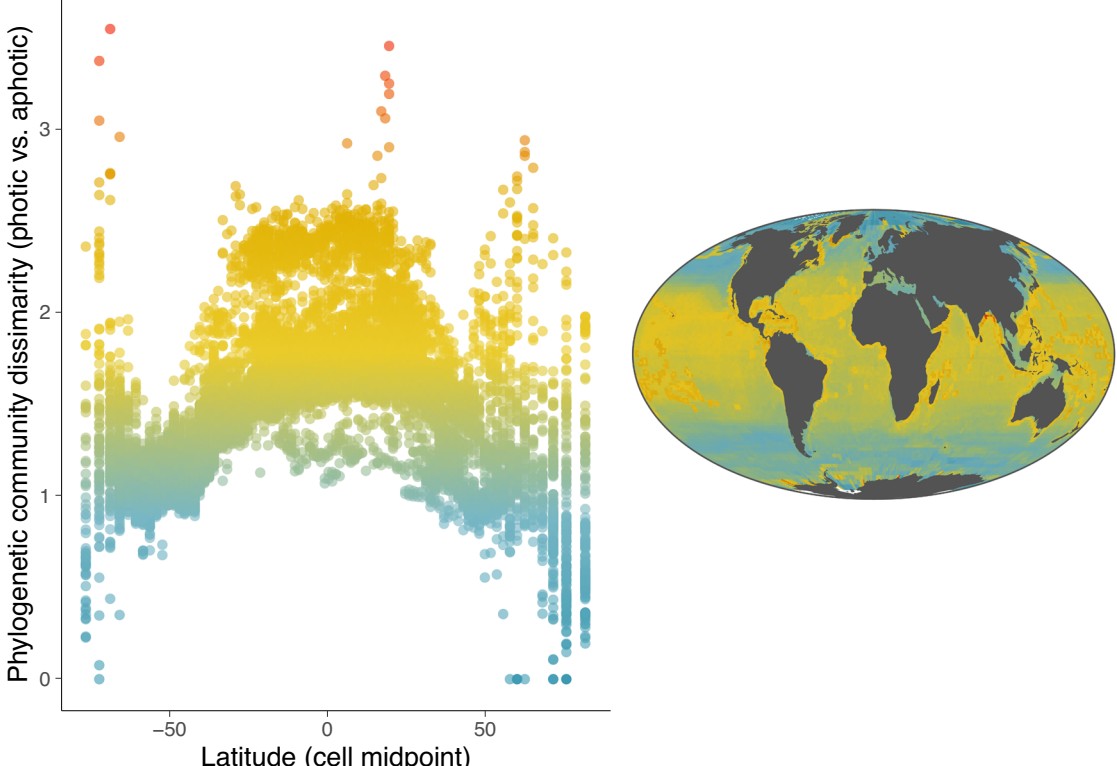

**Fig. 4 | High-latitude fish communities are more phylogenetically similar throughout the water column, whereas in the tropics, shallow and deep communities differ substantially.** Phylogenetic community dissimilarity between shallow and deep communities for each cell of a 150 × 150 km grid across the globe. Low values (blue) indicate greater phylogenetic similarity between shallow and deep fish communities. Source data are provided as a Source Data file.

both the mechanism and mode of speciation depend on the lineage, regional variables, and local conditions.

Indeed, depth provides a multitude of axes along which fishes can speciate, including feeding ecology, novel environmental regimes, habitat, and life history features, like life span[41,42]. For example, Antarctic icefishes speciate along the benthic-pelagic axis, with repeated invasions of pelagic, benthic, semipelagic and epibenthic niches and discrete vertical zonation by species[43]. This adaptability across the water column, particularly in the absence of a swim bladder[44], likely predisposes icefishes to diversify along the oceanic depth axis. Species also exhibit varied dietary niches[43], providing further opportunity for trophic differentiation along the depth gradient. By contrast, Pacific

rockfishes do not appear to exhibit such differentiation of traits related to diet[45]. Evidence from recently diverged sister species suggests that rockfishes are instead capitalizing on benthic rocky habitats, which are spatially discontinuous, in turn promoting speciation by limiting gene flow[46]. This mechanism may be compounded by the life history of rockfishes: like many fishes, larval stages occupy much shallower zones prior to settlement. Chance recruitment to different depth zones due to settlement in discontinuous habitats, like seamounts, or sea level fluctuations may help establish localized populations along the depth gradient[45]. In contrast to rockfishes, we speculate that the lack of a pelagic larval stage in snailfishes and eelpouts[47] may limit dispersal potential and increase opportunities for ecological speciation and

vicariant events. Hadal snailfishes are also known to have narrow depth ranges and species are endemic to specific trenches, which may serve as an engine for allopatric speciation[48]. Meanwhile, spatially overlapping eelpouts are known to stratify along the depth gradient, closely tracking bottom temperatures according to species-specific preferences[49], indicating that thermal niches, and thus depth, may play an important role in speciation in this group. Overall, we build on a foundation of studies that have suggested depth divergence may play a large role in the speciation of marine organisms that have few biogeographic barriers to dispersal[50–52]. Further investigation into the diversity of ecologies within and across depth zones and the physiological mechanisms for speciation in deep-sea fishes may provide resolution for some of the more enigmatic rapidly-diversifying groups, like snailfishes and flatfishes. Nonetheless, it is clear that diversification along the depth axis is a potent ingredient for speciation in marine fishes, pointing to a more generalizable effect of depth on evolutionary dynamics.

Just as intriguing as clades with high depth lability are those that have under-diversified across depth, a result primarily biased to tropical lineages. Wrasses, damselfishes, and gobies, all notoriously species-rich lineages, particularly on tropical coral reefs, transitioned along the depth gradient much less than expected, perhaps due to a lack of light availability, which many of these taxa rely on for integral aspects of their ecology (UV signaling, sexual selection, visual-based foraging, etc.). This result complements the surprisingly low rates of speciation also observed in these clades[9]. Likewise, there are other exceptions to the general patterns we observe between speciation rate, latitude, and depth lability. A few lower latitude clades, like flyingfishes (Exocoetidae) and gurnards (Triglidae), exhibit little depth variation despite rapid speciation rates. These clades are likely diversifying along other axes unrelated to the depth gradient, as both are relatively constrained by other aspects of their ecology. Flyingfishes are a surface-oriented group that are renowned for gliding considerable distances above the water to escape predation, and gurnards are a predominantly shallow benthic group (though we note this family was recently expanded to include the deeper-dwelling Peristediidae[53]) and spend a substantial portion of their adult life walking along the sea floor[54]. Additionally, a couple of closely related high-latitude clades—namely fatheads (Psychrolutidae) and poachers (Agonidae)—exhibit lower speciation rates. Of those, only poachers exhibit slightly elevated rates of depth evolution. It is unclear what has facilitated depth lability in these benthic armored fishes, but perhaps the lack of a swim bladder has relaxed constraints associated with pressure changes at depth[55]. Although our results affirm that depth is an important factor in driving speciation rates in some clades, diversification along the depth axis is not the sole mechanism by which rates of speciation increase in fishes. Clade-specific ecology and constraints are important features that can lead to differential evolutionary outcomes and need to be carefully considered for macroevolutionary studies.

## Why are depth transitions biased to the poles?

There is strong phylogenetic structure to observed latitudinal trends, as perciform fishes (i.e., an order of 'perch-like' fishes that contains over 40% of all bony fish species) account for 66.3% of high-latitude species[9], including four of the five most rapidly speciating clades. That spatial trends exhibit phylogenetic clustering extends beyond fishes[56], as oftentimes the highest ecological success of a broader taxonomic group is biased by a few extremely speciose clades, such as Passeriformes within birds and angiosperms within plants. Furthermore, high-latitude taxa across the Tree of Life tend to be phylogenetically nested within tropical lineages. Thus, we cannot entirely disentangle the effects of phylogeny from the relationship between niche lability and speciation. In fact, the story of fast speciation and depth evolution at high latitude is, by definition, also a story of high ecological success within a subset of fish diversity. Although this lack of phylogenetic

independence is an unavoidable feature of marine fish distributions, it more pertinently hints at some shift in evolutionary dynamics in these lineages. There may be some features of perciform fishes that facilitate both invasion of high latitudes and diversification along the depth gradient, as low latitude Perciformes (e.g., Serranidae, Platycephalidae, Triglidae, etc.) do not exhibit similar depth lability. For example, many deep-sea lineages, including deep-sea and polar snailfishes, have reduced bone density, which is thought enable neutral buoyancy at depth[48]. Reduced ossification is also a key feature of the Antarctic icefish adaptive radiation[57] and may play a role in the repeated deep-sea invasions observed in this lineage. Despite these insights, we still lack a thorough understanding of the factors that predispose particular lineages to diversify along the depth gradient (but see[58,59] for discussion on possible molecular mechanisms). Future research into the mechanisms that facilitate both invasion of high latitudes and diversification along the depth gradient in Perciformes may prove fruitful. Although high-latitude lineages are primarily limited to Perciformes, the proliferation of just a few lineages leaves a imprint on global biodiversity patterns[56].

In addition to phylogenetic bias, global patterns in diversification along the depth gradient may have a more mechanistic explanation. Our findings align with classic ecological theory proposed by Janzen[60], who posited that mountains pose a greater physiological barrier to dispersal for tropical species than temperate organisms. He reasoned that, because the tropics generally experience less seasonal variation in temperature than temperate regions, thermally specialized tropical species may be more likely to encounter regimes for which they are maladapted along an elevational gradient. Since its inception, Janzen's theory has been demonstrated across diverse taxa[61], cementing his seasonality hypothesis as an integral part of the ecological and evolutionary toolkit[34,62,63]. Although Janzen's theory was formulated around elevational gradients in terrestrial environments, analogous principles apply to depth gradients in marine systems[62] and studies have demonstrated that tropical aquatic organisms do encounter steeper thermal barriers to dispersal[64–66]. By contrast, diversification along the depth gradient might be comparatively easier for temperate and polar fishes, as they are already exposed to greater climatic heterogeneity and pronounced seasonal variation in the thermocline[67], resulting in more phylogenetically similar community structure across depth categories. High-latitude fishes can exhibit diel vertical migrations and seasonal depth changes[68], evidencing some plasticity that may further enable diversification along the depth axis. By contrast, tropical lineages confront a steeper temperature gradient, making depth transitions more physiologically challenging. Consistent with our findings, under this scenario we would expect to find both rapid rates of depth evolution and more similar community structure throughout the water column at higher latitudes. Marine fishes exhibit both faster rates of climatic niche[69] and range size evolution[70] at higher latitudes, suggesting that the selective pressures imposed by environmental factors may be relaxed in these systems. Indeed, higher latitudes have experienced severe climatic fluctuations on geological time scales, whereas tropical sea temperatures have been stable for 15my[71], factors that have undoubtedly shaped the thermal tolerances of organisms living in these regions[72]. Studies have demonstrated that diversity peaks at higher latitudes during warmer geological periods[3,73], evidencing the key role of temperature in governing global species diversification dynamics. Furthermore, these paleoclimatic changes, and the associated glaciation cycles, have been shown to correlate with the proliferation of Antarctic icefish lineages[74], eelpouts[75], and *Careproctus* snailfishes[47]. Glacial maxima intensify both thermal and physical barriers to diversification and can restrict life to a few open-ocean refugia with high productivity[74,76]. This results in periodic extinction of polar taxa, which creates opportunities for recolonization and rapid speciation during warmer eras, likely further facilitated by reduced thermal barriers at high latitudes[74,76].

In principle, Janzen's theory can be used as a lens for any physical factor that influences physiological performance[62,63]. In addition to temperature, many environmental parameters (e.g., dissolved $O_2$, pressure, light, salinity, etc.) vary with depth. Although temperature is a well-documented physiological barrier to organismal dispersal[6,77], there are likely a combination of oceanographic variables that impose strong constraints on species along the depth gradient[20]. For example, the combined effects of temperature and oxygen availability have been shown to impose metabolic constraints that shape global marine species distributions[78]. Notably, many of these oceanographic factors, including temperature and density, tend to be less vertically stratified at higher latitudes[25], which for greater mixing and primary productivity. Lower temperatures can also delay the onset of hyperbaric effects for organisms at depth[20]. In other words, the depth profiles of high-latitude environments tend to be more uniform across multiple dimensions than in the tropics, further lowering the proverbial mountain passes for high-latitude organisms. This concept may also explain why we find more transitions between deeper habitats across latitudes. The abyssal plains, which make up the majority of the deep sea, are notorious for relatively common, stable conditions—an absence of light, crushing pressures, and relatively uniform temperature and salinity[79,80]. Future work could examine the potential for this extreme environment to serve as a global conduit for fish diversification, reflecting the absence of physiological and biogeographic barriers to dispersal, once lineages are able to occupy habitats at extreme depth[81–83]. Overall, fish lineages have likely capitalized on the climatic variation and more homogenous water profile found at higher latitudes to readily diversify along the depth gradient. Thus, our results indicate that niche evolution may partially sustain the inverse latitudinal diversity gradient of speciation rates found in marine fishes.

### Trends in global community structuring

The latitudinal bias of depth transition rates produces differences in phylogenetic community structure between tropical and high-latitude regions. Whereas the phylogenetic composition of high-latitude fish communities is relatively similar throughout the water column, shallow tropical fish communities differ substantially from those at depth. Although it has been clear for decades that Antarctic fish communities are predominantly composed of icefishes[84] whereas tropical latitudes, particularly reefs, are dominated by wrasses, damselfishes, and gobies, here we quantify the sharp contrast in the diversity of global fish communities along a latitudinal gradient. These results align with other studies that have demonstrated the importance of environmental filtering for the community assembly of deep-sea octocorals[85,86] and the vertical zonation of fishes in Bermuda and New Zealand[18,87]. Insofar as phylogenetic relatedness reflects trait similarity[88], these results may suggest functional and/or physiological disparities between shallow and deep tropical fish communities, though this would require quantification as functional role is not always phylogenetically conserved. Although it is known that evolution can affect community assembly across broad temporal scales[89], we highlight the need for further research into the regional ecological and evolutionary mechanisms that govern community dynamics along the latitudinal gradient.

### Limitations

There are unavoidable sampling biases associated with studies on deep-sea organisms. Although observations of deep-sea communities commonly lead to discoveries of novel fish species, as would be expected in an environment that is under-sampled, we note that these new species are largely restricted to a few clades[90,91]. While we are likely underestimating the species richness of abyssal zones, these observations suggest that we have a decent understanding of deep-sea phylogenetic diversity at the larger taxonomic scales relevant to this study. If anything, further exploration of deep-sea fish communities

would likely reveal greater species diversification within the same focal families, increasing the estimated niche lability in these groups. Similarly, we recognize that we are likely underestimating depth lability in many of the deeper clades by virtue of our broad depth categories. Although finer categorization may yield more nuanced trends, such as those related to the diversity of habitats within each depth category, we settled on a more conservative approach here after considering the uncertainty in species depth estimates beyond broad classifications. Lastly, the scale and nature of this study necessitates use of the most comprehensive time-calibrated phylogeny available for fishes. Despite limitations in our ability to account for phylogenetic uncertainty, the redating and tip sensitivity analyses established that our findings are robust to alternative divergence time estimates and topological variation.

Here, we identify a latitudinal gradient for niche lability in marine fishes and find evidence that the depth axis is a crucial feature of diversification in high latitude, rapidly speciating lineages. We interpret these findings in light of classic ecological theory, unifying the latitudinal variation of oceanographic properties and the physiological tolerances of the species living there. Critically, global warming is destabilizing oceanic thermohaline circulation[92], threatening a cascade of effects including freezing much of the Northern Hemisphere and slowing global deep-water currents. These effects would strengthen abiotic gradients at high latitudes, with potentially severe implications for fishes that currently capitalize on the relative environmental similarity along the depth axis. Taken together, our results highlight a fundamental role for niche lability in shaping global patterns of marine fish diversity and underscore another potential vulnerability of polar ecosystems to changing environmental conditions.

## Methods
### Data acquisition

We obtained existing data on global species occurrence, as well as a time-calibrated phylogeny from Rabosky et al.[9], for a total of 4067 fish species across 286 families. This dataset includes information on species-specific geographic range extent, latitudinal midpoint of the geographic range, and tip-specific speciation rate. Briefly, the species-specific geographic ranges from Rabosky et al.[9] are based on available occurrence records and known environmental predictors, and were validated with primary literature and by consulting taxonomic experts[9]. Species were grouped into one of three depth categories designed to capture broad differences in environmental conditions (light, temperature, pressure, amount of wave disturbance, etc.) and consistent with common depth divisions in the literature[24,93]. The "shallow" depth category includes the well-lit epipelagic zone, which maintains a relatively uniform temperature profile, as well as the mixing layer (0–200 m; $n = 2548$ species). The "intermediate" category ranges from 200–1000 m ($n = 989$ species) and is equivalent to the mesopelagic zone, where temperature and ambient light decline, and density rapidly increases with depth. Lastly, the "deep" ocean is anything below 1000 m (bathypelagic and abyssopelagic zones) and contains 530 species in our dataset. Compared to shallower depths, this deep region of the ocean is characterized by a complete absence of ambient light and large expanses of relatively homogenous water (i.e., primarily the abyssal plains). We emphasize that this categorization system is intentionally broad and thus does not capture habitat diversity or the nuances of seasonal variation that are known to exist at all oceanic depths[93].

Species' depth information was extracted from Fishbase[94] using the R package, *rfishbase*[95], with values based on documented observations for adults and juveniles of species (Supplementary Data 1). Our data exclude larval stages, which can occupy different habitats and depths prior to settlement for some species. We checked the dataset for egregious errors and only altered the depths for two species of lanternfish, *Lampanyctus jordani* and *Gymnoscopelus hintonoides*,

which occupy much shallower depths, as detailed by the species description paragraph in Fishbase. Species were categorized based on their deepest observed occurrence and those without depth information were pruned from the dataset. We also corroborated depth data by confirming the ranges for a random sample of 100 species with other reliable sources (Supplementary Data 2). Categorizing species into broad depth zones was expected to largely mitigate inconsistencies in reported depth ranges, but we acknowledge that any approach has the potential for issues with data-deficient species and those near boundaries in depth category.

## Depth lability and speciation in focal clades

We sought to determine if transitions across the broad depth zones were greater than would be expected by chance within particular clades. Our expectation is that rapidly speciating fish clades[9] (e.g., Notothenioids and their allies, Pleuronectidae, Sebastidae, Zoarcidae, and Liparidae) will also exhibit greater lability along the depth axis than expected when compared to other fish clades. We first pruned our dataset to monophyletic clades ($n = 46$) with >20 species to ensure accurate estimates of clade-specific transition rates (Fig. S9, Table S2). We then reconstructed the evolution of depth using stochastic character maps (simmaps) implemented in *phytools*[96] (Fig. S10). Based on an initial log-Likelihood comparison of transition models (symmetric, equal, and all-rates-different), we produced 100 simmaps with "ARD" (all rates different) to model transition probabilities between depth states. For each simmap, we estimated how many transitions occurred within each of the 46 families since the most recent common ancestor (MRCA) of all extant species within the clade. We then calculated the median number of transitions for each family across the 100 simmaps. To account for the fact that older clades may exhibit more transitions simply by virtue of having more time to diversify along this axis, we estimated the rate of depth evolution for each clade by dividing the number of transitions by the age of the MRCA. To generate a null expectation, we simulated 100 discrete character datasets under the empirical transition matrix (Q) estimated from our simmaps using the sim.char function in the *geiger* package[97]. After generating 100 simmaps from these simulated datasets, we were then able to estimate the expected number of transitions, controlling for both clade age and the number of species within each family.

As this procedure necessitates discretizing depth, an inherently continuous variable, we repeated the above analysis with two modified depth categorization systems: (1) shallow: 0–100 m; intermediate: 100–900 m; deep: >900 and (2) shallow: 0–300 m; intermediate: 300–1100 m; deep: >1100 m. We also note that there is some variability in the depths at which environmental conditions shift due to geography, latitude, and seasonal differences[98,99]. This procedure allowed us to assess the robustness of our biological results to variation in assigned depth categories. We also recategorized species based on their median depth and reran the analysis with this modified criterion. We found that neither the depth categorization system nor the use of median depth substantially influences our findings or conclusions (Figs. S1 and S11).

To account for phylogenetic uncertainty, we used two different UCE phylogenies as references to re-date the Rabosky et al.[9] phylogeny: the phylogeny from Alfaro et al.[100] and the recently-published UCE phylogeny of Ghezelayagh et al.[33] For each tree, we used a congruification approach implemented in the R package 'geiger'[97] to identify shared nodes between each reference tree and the original Rabosky phylogeny and employed a penalized likelihood program (treePL) to recalibrate the original tree based on these shared nodes. Ultimately, this allowed us to retain the dense species sampling necessary for these analyses while accounting for variation in divergence time estimates. We re-ran all analyses with both redated phylogenies and the results are consistent with our previous findings (Figs. S2 and S3). To ensure our findings were robust to topological variation,

we also implemented a sensitivity analysis using the Rabsoky et al. phylogeny. We randomly resampled tips, ranging from 90–40% of the complete tree at 10% intervals 30 times each, for a total of 180 trees. We then repeated our analyses on this distribution of phylogenies, with remarkably stable results, even at just 40% of species sampled (Fig. S12).

Though there are existing methods to more directly assess the association between depth transitions and cladogenetic events[101,102], there are methodological concerns, particularly for large-scale studies with substantial rate heterogeneity[103,104], which make these analyses ill-suited for our purposes. Therefore, we performed both a standard and a phylogenetic regression (PGLS; phylogenetic generalized least squares) using the R package geomorph[105] to examine the relationship between the rate of depth evolution and speciation rate across all 46 clades. We assessed significance via a randomized residual permutation procedure (1000 replicates) in the package RRPP[106]. In addition to the BAMM speciation rates from Rabosky et al.[9], we repeated this analysis using two non-model-based approaches to estimate lineage-specific speciation rates (which were then averaged by clade). The first, node density, is a simple metric that quantifies the number of observed splits (nodes) through time and is unbiased with respect to branch length[107,108]. The second, the DR statistic[109], is similar to node density, but it is based on weighted inverse of phylogenetic branch lengths. We used these additional metrics to relax the assumption of homogeneous rates across all lineages and ensure rate estimates were not biased by pure-birth models[110].

## Depth and latitude evolution

To determine if transition rates between depth categories more generally differ by latitude across all fishes, we first generated a transition matrix across both depth and latitude. As this method requires discrete variables, we categorized latitude into tropical (0°–30°), temperate (30°–60°), and polar (60°–90°) regions, based on Earth's three major climatic zones. Species were categorized based on the location of their latitudinal centroid. We then used the fitMk function in *phytools* to fit Mk models, a model of discrete character evolution, with equal rates, symmetric rates, and all rates different transition models, and compared the fit of these different transition rate models using AIC. To ensure these results were robust to phylogenetic uncertainty, we repeated these analyses with the phylogeny from Ghezelayagh et al.[33] pruned to species in our dataset (the Alfaro et al. tree did not have adequate species sampling for this analysis).

Lastly, we estimated if latitude (e.g., tropical, temperate, and polar categories) affected the rate of depth evolution with a Bayesian approach using a state-dependent Brownian motion model of evolution. Though this analysis is slightly redundant with the procedure described above, it allowed us to retain depth as a continuous variable and, correspondingly, permitted a more robust estimate of the influence of latitude on depth evolution. Here, depth was calculated as the median depth occupied across the "DepthRangeShallow" and "DepthRangeDeep" variables obtained from Fishbase for each species. The MuSSCRat model[111] implemented in RevBayes[112] jointly estimates evolution of the discrete trait (latitude) and the continuous trait (median depth) while accounting for alternative (background) sources of rate variation, which has been shown to reduce the chance of erroneously associating a predictor variable with shifts in the evolutionary rate (i.e., reduced type-I error)[111]. We then estimated rates with an uncorrelated lognormal model in which the branch rates have no phylogenetic structure. This model has been shown to perform similarly to alternative models in which branch rates have phylogenetic structure (i.e., a random local clock[113]). We ran three Markov chain Monte Carlos (MCMCs) with 100,000 generations and a 10% burn-in with a prior expectation of 500 transitions between latitudinal regions, informed by previous simmap transition estimates. We repeated the analysis using two other priors on the number of state changes (400 and 600)

to determine the impact on posterior parameter estimates. The MCMCs were assessed using Tracer[114] to check for adequate Effective Sample Size.

## Global community comparisons

Based on our prediction of increased depth lability in high-latitude clades, we anticipated that the phylogenetic composition of communities should differ more between shallow and deep regions in the tropics than in temperate and polar regions. In other words, if lability along the depth gradient is higher at temperate and polar latitudes, then it is more likely that closely related species will reside at different depths, reducing the phylogenetic difference between shallow and deep communities. By contrast, low latitude species, under our expectations, would be less likely to transition along the depth gradient, resulting in stronger phylogenetic structuring between shallow and deep communities. We evaluated these predictions using the phylogenetic community dissimilarity (PCD) metric[115]. This metric is a combination of two distinct components: the first is a modification of Sorensen's index that removes the bias due to community size (PCDc), and the second evaluates the phylogenetic relationships of nonshared species between communities (PCDp). PCD is particularly advantageous because it does not require information about species abundance, which would not only be error-prone in a dataset of this size but also biased against more rarely captured deep-sea species. PCD = 0 when all species are shared between communities, whereas PCD = 1 implies that the communities are no more or less similar than communities selected at random from the total species pool. PCD values greater than 1 suggest that communities are more dissimilar than would be expected given a random assortment of species.

For this analysis, we categorized species as "shallow", "deep", or "both", depending on if they are found only above 200 m, only below 200 m, or both above and below 200 m, respectively. We used 200 m as the cut off given the dramatic shift in oxygen level, temperature, nutrients, and other environmental parameters around this depth, which may create strong physiological barriers to species dispersal[116]. We combined the depth information with species occurrence data collected by Rabosky et al.[9], which is constructed as a global grid with 150 × 150 km resolution and details the species present in each cell. We then pruned the phylogeny to the species present in each cell and evaluated PCD between shallow and deep communities in every marine cell, for a total of 22,736 cells globally. Due to computational limitations for some of the most species rich grid cells (<2%), we used an interpolation approach to estimate the PCD value as the average of the neighboring cells.

## Reporting summary

Further information on research design is available in the Nature Portfolio Reporting Summary linked to this article.

# Data availability

Data used from previously published sources is publicly available in the supplementary materials of Rabosky et al.[9] at https://doi.org/10.1038/s41586-018-0273-1. All other data supporting the findings of this study are available within the paper and its supplementary information files. Source data are provided with this paper.

# Code availability

R scripts used for the analyses described in this study are available at https://github.com/stfriedman/Depth-transitions-paper.

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

## Acknowledgements

We are grateful to Mike May and Edward Burress for guidance and computational support. Thank you to Peter Wainwright, Katherine Corn, Thomas Near, Michael Donoghue, and Elizabeth Miller, who provided advice and stimulating scientific discussion throughout this study. This work was funded by the G. Evelyn Hutchinson Environmental Post-doctoral Fellowship from Yale University.

## Author contributions

S.T.F. developed the concept of the paper and conducted data analysis. Both S.T.F. and M.M.M. contributed to the conceptualization and writing of the paper.

## Competing interests

The authors declare no competing interests.
