## [Peer Review File · Nature Communications]

A latitudinal gradient of deep-sea invasions for marine fishesREVIEWER COMMENTS

Reviewer #1 (Remarks to the Author):

This manuscript follows up on a recent finding that there is an inverse latitudinal gradient in speciation rates in marine fishes. In the current study the authors explore how depth may be shaping this pattern. I really liked the concept of this work, and it has fantastic analyses showing that diversification towards the poles is associated with depth. However, I have some major concerns and reservations that I think should be addressed concerning some of the analyses and interpretation.

Overall I feel like the paper fell short of its potential by staying overly broad and not providing more insight into biological hypotheses and interpretations of the data and results. It was also not organized well and jumped between topics, often without leaving much of an impression of what takeaway point I should have.

As I was reading this paper, I kept coming back to this quote: "In short, exceptional depth transitions are biased to a few, relatively closely related high-latitude fish clades, many of which are also known for rapid speciation rates." The introduction never quite gets to these clades that were identified previously and this is the first missing step, and neither the introduction nor the discussion solidify into a coherent framework.

It would have been nice to have the introduction focus on specific mechanisms and hypotheses that would be associated with depth in marine fishes. I would ask the authors to fundamentally revise this work to include clear hypotheses, and a discussion rooted in the biology and ecology of these organisms.

Another part of this problem seems to stem from the portrayal of deep-sea environments.

In the methods (line 98), the authors describe the deep-sea as having "little variation in oceanographic parameters, and large expanses of relatively homogenous water...". This was a historic view of deep-sea biodiversity and environments but has been found to certainly not be the case. There is high habitat heterogeneity within the deep sea as well as nutrient fluxes, etc. There are also seasonal pulses of phytodetritus that dramatically change communities, as well as high heterogeneity between environments such as the abyssal plain, continental margins, or sea mounts. At deeper depths, there is also the hadal zone to consider which is quite relevant to snailfishes. Considering these habitats, general aspects of population genetics, and the biology of deep-sea fishes in light of the clades of interest seems like the missing step to me. A focus on the biology here would help with the overall interpretation of the results beyond simply showing a pattern. For a review, please see Paulus 2021 "Shedding light on deep-sea biodiversity..".

The clades in this manuscript occupy a diverse range of habitats and this needs to be highlighted rather than binning them as "deep" or "intermediate." (Note by this I mean in terms of discussion and interpretation later, the analyses are fine).

The binning in the manuscript is usually used for light (euphotic, dysphotic, and aphotic), so I'm not sure this is capturing the broadest possible differences of variables such as pressure (line 91). This is fine, but should be phrased more carefully.

The authors should possibly consider the changes in oceanography that occur between latitudes more. For example, line 95: In presenting the intermediate category "where temperature and ambient light precipitously decline, and [water] density rapidly increases with depth." I'm not sure precipitous is warranted. Additionally, there is a latitudinal difference that is relevant to the manuscript. The thermocline and pycnocline create a barrier that prevents mixing between less dense surface water

and denser deeper water in tropical regions. However, polar regions have less density stratification, which allows for higher mixing and primary productivity. This could perhaps be of relevance when discussing the findings. There is one line mentioning this in the discussion, but this seems like it warrants far more attention in the set-up of the overall manuscript (see also below for Antarctic discussion and more nuance).

Discussion

The section on niche lability and speciation stays very broad and basically leaves a reader with not much to grab on to. This is a shame and reflects poor scholarship since some of the species discussed here rank among the more well-studied marine clades. For example, rockfish and the diversification of Antarctic icefish have been well-studied. Rather than just highlighting "depth" here repeatedly, discussing the environment and biology would allow the authors to contribute something more meaningful.

When discussing wrasses, damselfish, and gobies, perhaps the reason for a lack of transitions isn't simply a lack of corals at depth (note again that the larval forms are associated with deep-sea corals, also be careful in general of claims concerning coral distributions here (line 314-315)). What about light availability? Light perception, including UV signalling / sexual selection on color patterns, and visual foraging are key parts of the ecology of these clades.

Line 319: The discussion of Triglidae here is also not quite right, as this family has been revised to include Peristediidae (armored sea robins) that are predominately deeper dwelling. See Smith et al. 2018 Copeia. "Phylogeny and taxonomy of flatheads, scorpionfishes, sea robins..."

Line 330: This is confusing, wouldn't depth be a key part of the ecology of a lineage?

Line 350: reduced ossification is not unique to snailfish and Antarctic icefish, numerous deep-sea lineages have reduced ossification, this is likely a common convergent trait.

Line 352: There actually are hypotheses here. Considering the icefish, much of their radiation is quite recent and likely highly influenced by glacial dynamics like the rest of the Antarctic marine fauna. During times of glacial maxima primary productivity collapses at large spatial scales, glacial debris gets pushed down the continental slope, and life is restricted to a few polynyas around the continent (Thatje et al. 2008 Life Hung by a thread... Ecology). There have been a few papers invoking recovery from these extreme ecological conditions as the mechanism that allows communities to reassemble and promotes rapid speciation during periods of recovery (See Near et al 2012, Ancient Climate change PNAS...). For icefish, the molecular mechanism of depth change has been discussed (citation 71 in the current text), so wouldn't it be likely they are flowing into these newly created ecological opportunities that are further facilitated by the reduction in abiotic gradients?

Line 370: See above considering thermoclines in polar waters.... There are several manuscripts discussing the thermocline of the Antarctic (See Hattermann 2018 Antarctic Thermocline Dynamics, J. Physical Oceanography) and seasonal rhythms as well. It may also be worth mentioning some of the new data coming out of tracking polar fishes and the diel vertical migrations they undergo to reduce predation risks as well as changes in depth between the polar summer and winter. This seems quite relevant to consider, as these types of migrations provide the plasticity required to diversify along a depth axis.

Paragraph beginning on line 418: I agree with the authors, and think that rather than spending a sentence invoking that climate is important for birds and mammals, why not start with the fish here? It has been known for decades years that in the Antarctic, most of the biomass and diversity of the fish fauna is represented by icefishes. This contrasts pretty sharply with the diversity of shallow waters in tropical and temperate environments, and reflects the rapid extinction of a larger fish fauna

in the fossil record following the onset of polar conditions. All of this supports that climate change can shape global patterns of community diversity, and offers a stepping stone for discussing vacant niches and diversification along depth gradients.

Line 430: At this point the paragraph derails. Are high latitudes more functionally robust to environmental perturbations? I'm not sure this study supports such a claim, and this seems unlikely given the low thermal tolerance of many polar taxa. Colonization from another depth also isn't the only axis of replacement for tropical species (consider the wide diversity of marine habitats at similar depths in tropic or temperate latitudes), so this seems like a dangerous stretch. Maybe depth is simply a more important axis of within clade diversification at higher polar latitudes given the geologic/environmental backdrop?

I would also like to point out that the actual species within these clades can be functionally different, reflecting a very wide range of ecologies. Observing diversification in depth does not necessarily mean perfect functional conservation (the latter would need to be quantified).

Limitations: I appreciate this section, maybe also consider the coarse scales used in these analyses to characterize habitat as well as the data (see below).

Line 455: Again, invoking vulnerability here isn't quite justified since lineages can also move to other habitats over time and this was not studied. However, there is a missed opportunity here to discuss these results in light of forecasts of how climate change could impact the water column (see Boers 2021 Observation-based early warning signals for a collapse of the Atlantic Meridional Overturning Circulation Nature Climate Change; and similar).

Data

I'm also concerned with the depth data being used. The authors are mining data from fishbase, however, fishbase aggregates data and has quite large variances on depths for some species and very limited data for others. Maximum depth often does not reflect the depth at which adults (see next paragraph) commonly occur, as rogue individuals can be found out of the normal range. This is similar to finding East Asian birds in Europe within an occurrence database. The data is not shown or available (see section on data below) so it is difficult to evaluate how much of a problem this is, but I would imagine it could be quite pervasive in some taxa and that some manual curation of the dataset and checking of references will be necessary. This is analogous to dealing with spatial data to mitigate against rogue records.

Maximum depth may also not reflect the depth at which the species typically occurs, and it may be worth considering some alternate approaches to depth to assess whether this impacts analyses?

Along these lines, many of the species in this tree have pelagic larval stages that could give an artificial signature of deeper occurrence when using the maximum depth if these are included in some fishbase records.

The authors should note that deep-sea corals and other deep habitats act as nurseries for many larval forms of the tropical and temperate fish designated as "shallow" in these analyses. Are the authors only considering adult life stages in their analysis? This needs to be made explicit as many of the transitions discussed here occur within individuals of a species as part of their life cycle. If the authors are analyzing adults, the data needs to be assessed and curated to ensure consistency.

It should probably also be mentioned if the authors are considering depth only at one life stage in the manuscript (and thought of when discussing the evolution of water column usage).

Supplemental materials: Table S1 gives a summary by family but is limited to a description of the bins

(shallow, intermediate, deep). Can the authors include a table that includes summaries of the raw data and variance for these groups?

Figure S1: This is quite messy. The tip labels are not legible due to a small font size. Can this be cleaned up? Its ok to not show tip labels for this many species, but could the family names be outside of the tree in a border and maybe some corresponding alternating colored circles where the tip labels are now to show where families breaks occur? Some of the family names are not visible in the tree itself and it is hard to make sense of this figure relative to table S1.

Figure S2: One datapoint (Zoarcidae) seems like an outlier. What is the relationship when this point is removed (also in line 226)?

Figure S7A: A similar question for Notothenioid, what happens when this clade is removed?

Data and Code Availability: The authors link a repository for the tree file and speciation rates from an earlier paper and state all other data is publicly available and that they used no custom functions, just existing R packages. Given the repeatability crisis in science in general this is troublesome as anyone trying to repeat this study will have no way of verifying exact conditions. Even if the authors used existing functions, having the code in place to show how the resulting data was parsed and assembled, and arguments passed to functions is necessary.

Fishbase is not a static database. The data used in this study can be updated, at which point the study is no longer repeatable. Including the data and code in a repository with a DOI for this work seems like a trivial way to avoid such issues.

Figure 1: Can the lines between the null median and observed be removed? Also in figures throughout some clades have a little line going to a circle, others do not. Please remove.

Figure 4. I like this analysis, but is another result possible when you only have a few clades stratifying most of the water column in polar latitudes? This isn't a bad thing, but this figure could be set up better in the introduction and discussion to give another opportunity to highlight the contrast between higher and lower latitude clades.

Reviewer #2 (Remarks to the Author):

The study looks at the interplay between the latitudinal diversity gradient and the depth axis in explaining marine fish biodiversity patterns. To that end, the authors compiled depth, latitudinal data, and species-specific diversification rates for over 4,000 species, focusing mostly on 46 clades. They analyzed this dataset using state-of-the-art phylogenetic comparative methods applied to a recently proposed fish 'megatree.' They find that high latitude clades with fast speciation rates transition more frequently along the depth axis relative to those inhabiting tropical geographies, implying depth evolution lability is positively correlated with latitude. They also show that temperate or polar fish communities are more similar at different depths than those in tropical regions. While there are no new data associated with this manuscript (most datasets come from a previous publication; see below), I'm impressed by the originality of the study and its broadscale evolutionary and ecological implications.

I understand the value of using 'big data' and cutting-edge analytical tools to provide explicit tests of hypotheses at large scales. A major concern, however, is that the phylogenetic 'megatree' (from Rabosky et al.) informing the comparative analyses has multiple quality issues, which may ultimately undermine their results and interpretations. I realize that this tree has become extensively used for

many large-scale phylogenetic comparative studies recently. But I encourage the authors to take a careful look at the relationships depicted by this tree: they will find that ~15% of the families with >1 species that are indisputably monophyletic based on both molecules and morphology fail to pass a simple monophyly test (the total families with >1 sp. that are not monophyletic is ~27%, but that figure includes families that are evidently not monophyletic based on molecular evidence). And that is not taking into account other relationships (there are many 'good' genera recovered as grossly polyphyletic in this tree) or even divergence times (compared to other fish trees out there, this tree comprises multiple outlier age estimates). How can we trust the results of their macroevolutionary inferences if the relationships and ages in the tree that provides the basis for all downstream analyses are unreliable?

Ultimately, I believe evolutionary biologists using phylogenetic comparative methods must raise the 'quality bar' by accounting for potential sources of error in comparative inferences, arising not only from tree uncertainty but also from (ecological) data error (e.g., Silvestro et al., 2015). One solution I see is to build or use other trees (e.g., Open Tree of Life or TimeTree of Life) and assess the extent to which phylogenetic error may have affected their comparative inferences, including (critically) estimates of tip-specific speciation rates. It is also key to revisit the inverse latitudinal gradient identified by the previous study, which underpins all of their analyses. Species-level ecological data obtained from large-scale databases (FishBase) or secondary sources (Rabosky et al.) can also be unreliable. This can be validated by using depth and latitude information for a random selection of species in their dataset extracted from the primary literature.

Other points:

Global community structure comparisons: "Therefore, in tropical latitudes the shallow fish community strongly differs from the deeper fish community, reflecting limited diversification along the depth axis" & "High-latitude communities appear more phylogenetically homogenous along the depth axis, likely reflecting repeated diversification across depth categories." The PCD metric used for this analysis simply compares phylogenetic similarity/dissimilarity between communities at different depths within each latitude. Any observed dissimilarities will be the result of two factors: transition and diversification rates. Please clarify how the PCD metric can tease these factors apart.

"We produced 100 simmaps with "ARD" (all rates different) to model transition probabilities between depth states based on an initial log-Likelihood comparison from a sample of reconstructions." Why not first compare the fit of different models and then chose the best fit model(s) to reconstruct depth?

"We then used the fitMk function in phytools 37 to fit Mk models 47,48, a model of discrete character evolution, with equal rates, symmetric rates, and all rates different transition models, and compared the fit of these different transition rate models using AIC." Related to the above point, why is this analysis done for latitude but not depth?

The information presented in the M&M section under "Latitude and Depth Transitions" is poorly organized. For example, it's unclear why estimates of diversification rates and their relationship with depth go in this section. Without a more structured section layout, it's hard to keep track of all the different analyses and how they are connected.

Figure 1. "(green: greater than expectation, grey: within expectation, yellow: below expectation)" For ease of interpretation, please embed a symbol box with this info. onto the figure.

Figure 4 & elsewhere. "High latitude fish communities are more phylogenetically similar throughout the water column, whereas in the tropics, shallow and deep communities differ substantially." While this interpretation is indeed supported by the plot, there are also many more outliers (greater dispersion of data points) in high vs. low latitudes, particularly towards the left (negative latitude values). I would like to see an interpretation for this.

Figures S2, S3 & S7 are in my opinion too important to have them buried in the SM. Please consider moving them to the main text (if display items are at the limit, multiple panels can always be combined into a single fig.)

"We demonstrate that clades that with the highest," Typo.

"This pattern also extended to several other families, such as lanternfishes (Myctophidae), jack (Carangidae)." Should be 'jacks.'

Reviewer #3 (Remarks to the Author):

This manuscript presents a study that finds evolutionary transitions in oceanic depth contribute to the recently described inverse latitudinal gradient in ray-finned fish speciation rates. This counterintuitive pattern—highest rates of speciation at high latitudes despite highest diversity at low latitudes—defies classic ideas about the tropics as a center for diversification, and the identification of a potential mechanistic explanation is exciting. The authors show that rapidly speciating, high latitude clades exhibit exceptionally high rates of transition between depth zones supporting the predicted association between depth and speciation rate at high latitudes. More generally, they also demonstrate that transitions between depth zones occur more frequently at higher latitudes compared to the tropics and highlight a central role for evolutionary lability of depth in the diversification of fishes. The analyses are thorough, the manuscript is well written, and the figures provide especially clear illustrations of the central results. Despite this promise, however, I have several conceptual and methodological concerns.

1) I would like the authors to more clearly explain the hypothesized dynamics of diversification with depth transitions. It seems that there are two possible mechanisms, and I was unclear on which was thought to be at work. First, shifts in depth might prompt speciation because adaptation to novel environmental conditions (particularly in the deep ocean) leads to reproductive isolation. So, depth transitions would tend to be associated with speciation events, and a clade that has experienced more depth transitions would also have had more speciation. A second mechanism is that transitions to deep water environments provide ecological opportunity. In this case, depth transitions would be followed by niche divergence and speciation within deeper water environments. The authors suggest both mechanisms as possible drivers of speciation (lines 66-71), but I think more explicit differentiation between mechanism and predicted evolutionary outcomes is required. As I describe in my next comment, character state-associated diversification models could be used to disentangle these different effects and could potentially provide novel insights, such as why some lineages with depth lability exhibit high speciation rates but others do not.

2) The authors should consider applying models that simultaneously account for character state-dependent diversification in their estimation of depth transition rates. Estimation of character state transitions that take the phylogeny as given can be biased when that character influences diversification (Maddison 2006). For this reason, I am somewhat concerned about the robustness of the result that depth transition rates tend to be higher in rapidly speciating and/or high latitude clades. Depending on the hypothesized dynamics of depth and speciation (see previous comment), it may be more appropriate and insightful to use a model like BiSSE (Maddison et al. 2007), MuSSE (FitzJohn 2012), or HiSSE (Beaulieu and O'Meara 2016) if depth spurs speciation by providing ecological opportunity, or BiSSE-ness (Magnuson-Ford et al. 2012) if depth transitions are associated with speciation events.

3) Related to the two comments above, I am also somewhat concerned by the use of a clade-based approach for investigating diversification dynamics. The sampling units for the first part of the analysis are named clades, but this partitioning of the phylogeny is somewhat arbitrary. This approach may

obscure the role of depth as other unconsidered factors that are shared within a clade can also contribute to diversification. While the association between depth transitions and speciation rate is repeated across multiple high latitude clades, there are a couple of results that complicate inferences about a causal role for depth. First, the clades that drive the relationship are relatively closely related; when phylogeny is accounted for in the regression of speciation and depth transition rates, the relationship is no longer significant (lines 222-224). To be fair, the authors are aware of this complication and address it to some extent in the Discussion. The second complicating result is that several additional clades with high rates of depth transitions do not show high speciation rates. Combined, these results suggest that the effect of depth is perhaps not pervasive but rather context or lineage dependent. I recommend the use of state-based diversification methods (see comment 2) may clarify the generality of the depth's effect on diversification across lineages.

4) I recommend further evaluation of the robustness of these results to phylogenetic uncertainty. While I applaud the authors consideration of alternative categorizations of depth and analytical approaches, the phylogeny is treated as known. However, error in estimation of divergence times and poor resolution in some nodes may have considerable influence on estimates of evolutionary rates (of depth and latitude transitions as well as speciation) and therefore influence inferred relationships between depth and latitude transitions and speciation. Repeating analyses on a sample of plausible phylogenetic trees could help address this issue, though I recognize some of the methods used in this study are computationally demanding and this strategy may be time intensive. Nevertheless, some assessment of the role of phylogenetic uncertainty is necessary to demonstrate that key results are generalizable beyond this phylogenetic estimate.

5) A final, minor comment is that I would like the authors to justify the use of different ways of representing species depth in different analyses. Estimates of rates of depth transitions are based on species deepest occurrence (lines 101-103) but tests of association between latitude and rates of depth evolution involve species' median depths (lines 161-163). Also, do these alternative descriptions of depth alter interpretation at all?

REVIEWER COMMENTS

Reviewer #1 (Remarks to the Author):

This manuscript follows up on a recent finding that there is an inverse latitudinal gradient in speciation rates in marine fishes. In the current study the authors explore how depth may be shaping this pattern. I really liked the concept of this work, and it has fantastic analyses showing that diversification towards the poles is associated with depth. However, I have some major concerns and reservations that I think should be addressed concerning some of the analyses and interpretation.

Overall I feel like the paper fell short of its potential by staying overly broad and not providing more insight into biological hypotheses and interpretations of the data and results. It was also not organized well and jumped between topics, often without leaving much of an impression of what takeaway point I should have.

We appreciate the thorough and helpful feedback from this reviewer. In light of their constructive comments, we substantially rewrote and reorganized portions of the manuscript, with special attention to the introduction and discussion to improve clarity, detail, and flow. In this vein, we have also expanded discussion of the ecology and biology of the fishes in our study, so as to link broadscale pattern to mechanistic underpinnings more meaningfully. We believe that this additional detail and clarity greatly strengthen the study and we hope that the reviewer finds the revised version of the manuscript substantially improved in these respects.

As I was reading this paper, I kept coming back to this quote: "In short, exceptional depth transitions are biased to a few, relatively closely related high-latitude fish clades, many of which are also known for rapid speciation rates." The introduction never quite gets to these clades that were identified previously and this is the first missing step, and neither the introduction nor the discussion solidify into a coherent framework.

We agree and now identify the five high-latitude lineages that are the focus of this study in the second paragraph of the introduction. As mentioned above, we have also substantially restructured the introduction and discussion in an effort to improve the clarity and to solidify the framework of this study.

It would have been nice to have the introduction focus on specific mechanisms and hypotheses that would be associated with depth in marine fishes. I would ask the authors to fundamentally revise this work to include clear hypotheses, and a discussion rooted in the biology and ecology of these organisms.

We agree. In our revision, we expanded the introduction to include specific hypotheses, which serve to better set up our analyses and the organization of the paper. We have likewise made significant changes to the discussion to focus more on the biology and ecology of the five focal lineages, as well as inferences underlying the mechanisms of diversification along the depth axis grounded in the literature on these lineages.

Another part of this problem seems to stem from the portrayal of deep-sea environments.

In the methods (line 98), the authors describe the deep-sea as having "little variation in oceanographic parameters, and large expanses of relatively homogenous water...". This was a historic view of deep-sea biodiversity and environments but has been found to certainly not be the case. There is high habitat heterogeneity within the deep sea as well as nutrient fluxes, etc. There are also seasonal pulses of phytodetritus that dramatically change communities, as well as high heterogeneity between environments such as the abyssal plain, continental margins, or sea mounts. At deeper depths, there is also the hadal zone to consider which is quite relevant to snailfishes. Considering these habitats, general aspects of population genetics, and the biology of deep-sea fishes in light of the clades of interest seems like the

missing step to me. A focus on the biology here would help with the overall interpretation of the results beyond simply showing a pattern. For a review, please see Paulus 2021 "Shedding light on deep-sea biodiversity..".

We acknowledge that our description of the deep sea is an oversimplification of the habitat diversity present. Although there are hotspots of diversity around complex habitats like hydrothermal vents and seamounts in the deep sea, they are incredibly rare in the grand scheme of deep-sea habitats. Over 50% of the earth's surface is composed of abyssal plains with relatively homogenous conditions from an oceanographic perspective (Paulus 2021). Especially when compared to shallower habitats (<200m), the deep sea contains much less habitat diversity and variation in abiotic parameters, which is what we were attempting to emphasize in this portion of the manuscript. Nonetheless, we appreciate the reviewer's point and recognize that there is oceanographic variation in the deep-sea that we had previously glossed over in the text. We now have added a caveat to our description of the deep-sea in the methods section: "We emphasize that this categorization system is intentionally broad and thus does not capture the localized habitat diversity or the nuances of seasonal variation that are known to exist at all oceanic depths (Paulus 2021)."

The clades in this manuscript occupy a diverse range of habitats and this needs to be highlighted rather than binning them as "deep" or "intermediate." (Note by this I mean in terms of discussion and interpretation later, the analyses are fine).

We completely agree that the habitat diversity of these oceanic depth categories is important. Though investigating such habitat diversity is beyond the scope of this study, the reviewer is correct that habitat likely plays a role in speciation along the depth axis. We now mention the breadth of habitats found within each of these depth categories at various points in the manuscript (lines 115; 360-364; 375, 531).

The binning in the manuscript is usually used for light (euphotic, dysphotic, and aphotic), so I'm not sure this is capturing the broadest possible differences of variables such as pressure (line 91). This is fine, but should be phrased more carefully.

The sentence now reads: "Species were grouped into one of three depth categories designed to capture broad differences in environmental conditions (light, temperature, pressure, amount of wave disturbance, etc.) and in line with common depth divisions found in the literature."

The authors should possibly consider the changes in oceanography that occur between latitudes more. For example, line 95: In presenting the intermediate category "where temperature and ambient light precipitously decline, and [water] density rapidly increases with depth." I'm not sure precipitous is warranted. Additionally, there is a latitudinal difference that is relevant to the manuscript. The thermocline and pycnocline create a barrier that prevents mixing between less dense surface water and denser deeper water in tropical regions. However, polar regions have less density stratification, which allows for higher mixing and primary productivity. This could perhaps be of relevance when discussing the findings. There is one line mentioning this in the discussion, but this seems like it warrants far more attention in the set-up of the overall manuscript (see also below for Antarctic discussion and more nuance).

We have removed the word "precipitously" from this sentence.

We have also revised the introduction to more clearly emphasize that the high latitude environments are less vertically stratified with respect to the environmental parameters (and specifically mention temperature and density). The introduction now reads: "Inevitably, these environmental parameters vary substantially along the depth gradient from shallow seas to the abyssal plains, but there is also latitudinal structuring to this vertical stratification, such that these parameters, like temperature and density, are less stratified at high latitudes and may reduce barriers to depth diversification."

Additionally, there are two paragraphs (starting on line 443) in the discussion that detail how different environmental parameters vary with both depth and latitude and how this may facilitate diversification across the depth gradient at high latitudes, driving the overall patterns we find in this study. We now explicitly mention the latitudinal variation in mixing and primary productivity in these paragraphs (line 487).

Discussion

The section on niche lability and speciation stays very broad and basically leaves a reader with not much to grab on to. This is a shame and reflects poor scholarship since some of the species discussed here rank among the more well-studied marine clades. For example, rockfish and the diversification of Antarctic icefish have been well-studied. Rather than just highlighting “depth” here repeatedly, discussing the environment and biology would allow the authors to contribute something more meaningful.

We completely agree with this critique and have paid careful attention in our revision to meaningfully link the broadscale patterns we observe with the biology of the organisms. We have added another paragraph to the discussion which details more of the environment and biology of the different clades we discuss in this manuscript. We also highlight some of the clades in which diversification with depth has been explored, drawing a solid connection to the underlying ecology of some of the better-studied clades and the potential for niche lability along the depth gradient.

When discussing wrasses, damselfish, and gobies, perhaps the reason for a lack of transitions isn't simply a lack of corals at depth (note again that the larval forms are associated with deep-sea corals, also be careful in general of claims concerning coral distributions here (line 314-315)). What about light availability? Light perception, including UV signalling / sexual selection on color patterns, and visual foraging are key parts of the ecology of these clades.

This is a good point. We have removed the part of this sentence that references corals and now detail the ways in which many of these taxa rely on light perception.

Line 319: The discussion of Triglidae here is also not quite right, as this family has been revised to include Peristediidae (armored sea robins) that are predominately deeper dwelling. See Smith et al. 2018 Copeia. “Phylogeny and taxonomy of flatheads, scorpionfishes, sea robins...”

We now make note of this in the text (line 405). There are no species in our analysis from Peristediidae.

Line 330: This is confusing, wouldn't depth be a key part of the ecology of a lineage?

Agree. We have changed the sentence to read “...constrained by other aspects of their ecology”.

Line 350: reduced ossification is not unique to snailfish and Antarctic icefish, numerous deep-sea lineages have reduced ossification, this is likely a common convergent trait.

We now specify that this trait is not limited to snailfishes and icefishes, but rather a generalizable trait across many deep-sea lineages.

Line 352: There actually are hypotheses here. Considering the icefish, much of their radiation is quite recent and likely highly influenced by glacial dynamics like the rest of the Antarctic marine fauna. During times of glacial maxima primary productivity collapses at large spatial scales, glacial debris gets pushed down the continental slope, and life is restricted to a few polynyas around the continent (Thatje et al. 2008 Life Hung by a thread... Ecology). There have been a few papers invoking recovery from these extreme ecological conditions as the mechanism that allows communities to reassemble and promotes rapid speciation during periods of recovery (See Near et al 2012, Ancient Climate change PNAS...). For icefish, the molecular mechanism of depth change has been discussed (citation 71 in the current text), so

wouldn't it be likely they are flowing into these newly created ecological opportunities that are further facilitated by the reduction in abiotic gradients?

This is true and we agree it is very pertinent to mention these studies in our discussion. We have now expanded the paragraph to include a discussion of how glaciation dynamics may contribute to diversification dynamics at high latitudes (lines 470-476).

Line 370: See above considering thermoclines in polar waters.... There are several manuscripts discussing the thermocline of the Antarctic (See Hattermann 2018 Antarctic Thermocline Dynamics, J. Physical Oceanography) and seasonal rhythms as well. It may also be worth mentioning some of the new data coming out of tracking polar fishes and the diel vertical migrations they undergo to reduce predation risks as well as changes in depth between the polar summer and winter. This seems quite relevant to consider, as these types of migrations provide the plasticity required to diversify along a depth axis.

We thank the reviewer for the suggestion and have added both the citation and a brief discussion of diel vertical migration in the discussion (line 458). Unfortunately, we are not aware of the literature on tracking of polar fishes that the reviewer suggests including and were unable to locate recent relevant papers on this topic.

Paragraph beginning on line 418: I agree with the authors, and think that rather than spending a sentence invoking that climate is important for birds and mammals, why not start with the fish here? It has been known for decades years that in the Antarctic, most of the biomass and diversity of the fish fauna is represented by icefishes. This contrasts pretty sharply with the diversity of shallow waters in tropical and temperate environments, and reflects the rapid extinction of a larger fish fauna in the fossil record following the onset of polar conditions. All of this supports that climate change can shape global patterns of community diversity, and offers a stepping stone for discussing vacant niches and diversification along depth gradients.

We have removed the sentence that references mammals and birds in this paragraph. We have also added a sentence that details the qualitative evidence of community changes along the latitudinal gradient, including detailing the predominate fish clades in high vs. low latitudes.

Line 430: At this point the paragraph derails. Are high latitudes more functionally robust to environmental perturbations? I'm not sure this study supports such a claim, and this seems unlikely given the low thermal tolerance of many polar taxa. Colonization from another depth also isn't the only axis of replacement for tropical species (consider the wide diversity of marine habitats at similar depths in tropic or temperate latitudes), so this seems like a dangerous stretch. Maybe depth is simply a more important axis of within clade diversification at higher polar latitudes given the geologic/environmental backdrop?

We agree and have removed this part of the discussion.

I would also like to point out that the actual species within these clades can be functionally different, reflecting a very wide range of ecologies. Observing diversification in depth does not necessarily mean perfect functional conservation (the latter would need to be quantified).

We agree with this statement and have revised the text to clarify that further research would be needed to substantiate this hypothesis.

Limitations: I appreciate this section, maybe also consider the coarse scales used in these analyses to characterize habitat as well as the data (see below).

We now note that our broad depth categories obscure finer scale trends like those associated with particular habitats.

Line 455: Again, invoking vulnerability here isn't quite justified since lineages can also move to other

habitats over time and this was not studied. However, there is a missed opportunity here to discuss these results in light of forecasts of how climate change could impact the water column (see Boers 2021 Observation-based early warning signals for a collapse of the Atlantic Meridional Overturning Circulation Nature Climate Change; and similar).

This is an excellent point, and we appreciate the recommendation. We have removed the reference to the vulnerability of species and now discuss how climate change may alter the ocean circulation dynamics which sustain the latitudinal trends in environmental parameters that we invoke throughout this study.

Data

I'm also concerned with the depth data being used. The authors are mining data from fishbase, however, fishbase aggregates data and has quite large variances on depths for some species and very limited data for others. Maximum depth often does not reflect the depth at which adults (see next paragraph) commonly occur, as rogue individuals can be found out of the normal range. This is similar to finding East Asian birds in Europe within an occurrence database. The data is not shown or available (see section on data below) so it is difficult to evaluate how much of a problem this is, but I would imagine it could be quite pervasive in some taxa and that some manual curation of the dataset and checking of references will be necessary. This is analogous to dealing with spatial data to mitigate against rogue records.

We now include a supplementary table that details the raw information from Fishbase, including the maximum and minimum depth recorded for each species. We acknowledge that there is some potential for maximum depth to be erroneous; however, we suspect that shallow water observations are more prone to recordings of rogue individuals. This is because fishing is biased towards relatively shallower depths, increasing the likelihood of observing any rogue individuals. Nevertheless, we relied on a coarse categorization system for the depth zones to largely mitigate inconsistencies in reported depths. We also emphasize that, considering the size of this dataset and the thousands of species present, even if there are observations of rogue individuals for a handful of species, it would not substantially influence our findings. The only reasonable way the underlying habitat data would influence this study is if it were biased in some systematic way, which we have no reason to believe is the case here. However, we agree that it is important to be explicit about our inability to account for uncertainty with respect to depth and we now detail this in the revised version of the manuscript (line 553).

Furthermore, in response to this comment, we reran our analyses with species recategorized based on median observed depth (Fig. S4). Our primary conclusions remain unchanged with these new analyses. Overall, this lends additional support that our findings are robust to variation in depth categorization methods. Lastly, in response to another reviewer's similar concerns, we corroborated depth data using other sources for a random sample of 100 species from our dataset (Table S2). The depth ranges largely matched those reported by Fishbase and the few with minor discrepancies ultimately did not alter the species' depth category. The one exception is *Lyconus brachycolus*, which had a Fishbase maximum depth of 997m, while other sources reported 1,000m. Although it is likely due to rounding error, this would recategorize this species from 'intermediate' to 'deep' in our dataset. We acknowledge that any large-scale macroevolutionary approach, including our own, is vulnerable to sampling error, particularly in rare species and those near depth zone boundaries and we now clarify these limitations in the methods (lines 127-129).

Maximum depth may also not reflect the depth at which the species typically occurs, and it may be worth considering some alternate approaches to depth to assess whether this impacts analyses?

We have done so; please see our response to the above comment.

Along these lines, many of the species in this tree have pelagic larval stages that could give an artificial signature of deeper occurrence when using the maximum depth if these are included in some fishbase records.

This is a good point. We checked and according to the Fishbase website the data is "the depth range (in m) reported for juveniles and adults (but not larvae)". Therefore, we have no expectation that depths

recorded for pelagic larval stages would bias our data. We have now explicitly stated this in the methods.

The authors should note that deep-sea corals and other deep habitats act as nurseries for many larval forms of the tropical and temperate fish designated as “shallow” in these analyses. Are the authors only considering adult life stages in their analysis? This needs to be made explicit as many of the transitions discussed here occur within individuals of a species as part of their life cycle. If the authors are analyzing adults, the data needs to be assessed and curated to ensure consistency.

Please see our responses above, as information on larval depth is not curated by Fishbase.

It should probably also be mentioned if the authors are considering depth only at one life stage in the manuscript (and thought of when discussing the evolution of water column usage).

This is an excellent point, and we now make note that we are not considering larval stages here.

Supplemental materials: Table S1 gives a summary by family but is limited to a description of the bins (shallow, intermediate, deep). Can the authors include a table that includes summaries of the raw data and variance for these groups?

We have added a new supplemental figure that details the spread of depth data within each family (Fig. S1). We also include the raw depth data for each species in Table S1.

Figure S1: This is quite messy. The tip labels are not legible due to a small font size. Can this be cleaned up? Its ok to not show tip labels for this many species, but could the family names be outside of the tree in a border and maybe some corresponding alternating colored circles where the tip labels are now to show where families breaks occur? Some of the family names are not visible in the tree itself and it is hard to make sense of this figure relative to table S1.

This figure is high resolution such that zooming in on different parts of the tree makes the tip labels clearly legible. We feel that including the tip labels is important to allow for full interpretation of our data, such as the species included in each clade and the habitat reconstruction leading to that tip. In response to this feedback, we now display the family names on the outside of the tree in a border, as suggested.

Figure S2: One datapoint (Zoarcidae) seems like an outlier. What is the relationship when this point is removed (also in line 226)?

The results are still significant ($F = 5.73$, $p = 0.016$).

Figure S7A: A similar question for Notothenioid, what happens when this clade is removed?

The results are even stronger without Notothenioids ($F = 32.88$, $p < 8.896e-07$).

Data and Code Availability: The authors link a repository for the tree file and speciation rates from an earlier paper and state all other data is publicly available and that they used no custom functions, just existing R packages. Given the repeatability crisis in science in general this is troublesome as anyone trying to repeat this study will have no way of verifying exact conditions. Even if the authors used existing functions, having the code in place to show how the resulting data was parsed and assembled, and arguments passed to functions is necessary.

An excellent point – we completely agree and strive for transparency in our work. The code to carry out all analyses and regenerate all figures will be publicly available on the associated Dryad repository upon acceptance.

Fishbase is not a static database. The data used in this study can be updated, at which point the study is

no longer repeatable. Including the data and code in a repository with a DOI for this work seems like a trivial way to avoid such issues.

We also agree with this point and now include the table of raw depth data in the supplemental materials (Table S1).

Figure 1: Can the lines between the null median and observed be removed? Also in figures throughout some clades have a little line going to a circle, others do not. Please remove.

We appreciate the feedback but feel the lines between the null median and the observed aid in interpretation for the figure and prefer to keep them as is. The lines in the other figures are to label points located in densely populated regions of the figure with the family name. These labels (and the associated lines) are automatically generated and positioned for clarity and readability of the figure. We have modified them in the updated version of the figure to remove any inconsistencies.

Figure 4. I like this analysis, but is another result possible when you only have a few clades stratifying most of the water column in polar latitudes? This isn't a bad thing, but this figure could be set up better in the introduction and discussion to give another opportunity to highlight the contrast between higher and lower latitude clades.

We now include an explicit hypothesis in the introduction relating to this analysis and have expanded the setup for this expectation in the introduction: "...niche lability could alter community composition in a predictable way such that shallow and deep communities of fishes may not substantially differ (phylogenetically) in regions that are dominated by a few clades which also exhibit elevated depth variation."

Reviewer #2 (Remarks to the Author):

The study looks at the interplay between the latitudinal diversity gradient and the depth axis in explaining marine fish biodiversity patterns. To that end, the authors compiled depth, latitudinal data, and species-specific diversification rates for over 4,000 species, focusing mostly on 46 clades. They analyzed this dataset using state-of-the-art phylogenetic comparative methods applied to a recently proposed fish 'megatree.' They find that high latitude clades with fast speciation rates transition more frequently along the depth axis relative to those inhabiting tropical geographies, implying depth evolution lability is positively correlated with latitude. They also show that temperate or polar fish communities are more similar at different depths than those in tropical regions. While there are no new data associated with this manuscript (most datasets come from a previous publication; see below), I'm impressed by the originality of the study and its broadscale evolutionary and ecological implications.

We thank the reviewer for their thoughtful and thorough feedback and have made every effort to address their concerns. Please see our specific comments to each point below.

I understand the value of using 'big data' and cutting-edge analytical tools to provide explicit tests of hypotheses at large scales. A major concern, however, is that the phylogenetic 'megatree' (from Rabosky et al.) informing the comparative analyses has multiple quality issues, which may ultimately undermine their results and interpretations. I realize that this tree has become extensively used for many large-scale phylogenetic comparative studies recently. But I encourage the authors to take a careful look at the relationships depicted by this tree: they will find that ~15% of the families with >1 species that are indisputably monophyletic based on both molecules and morphology fail to pass a simple monophyly test (the total families with >1 sp. that are not monophyletic is ~27%, but that figure includes families that are evidently not monophyletic based on molecular evidence). And that is not taking into account other relationships (there are many 'good' genera recovered as grossly polyphyletic in this tree) or even divergence times (compared to other fish trees out there, this tree comprises multiple outlier age estimates). How can we trust the results of their macroevolutionary inferences if the relationships and ages in the tree that provides the basis for all downstream analyses are unreliable?

The reviewer brings up an important point. In response to this comment and another in a similar vein from reviewer 3, we recognize that we have assumed the phylogeny as known here and that there are concerns about phylogenetic uncertainty in terms of the topology and dating of nodes. We appreciate this feedback and agree that, ideally, macroevolutionary studies should run analyses on a posterior distribution of phylogenies to account for uncertainties. Unfortunately, due to the massive phylogenetic scale of this work, there are no other existing, published phylogenies at our disposal that are suitable for these purposes. In response to this comment, we tried running analyses on every other published ray-finned fish megatree (Betancur-R. et al. 2013; Alfaro et al. 2018; Hughes et al. 2018; Ghezelayagh et al. 2022), and, without fail, none had dense enough sampling for our purposes. For our study, we have limited our analyses to clades with at least 20 species present to ensure we are accurately capturing the number of transitions between habitats in each clade. Without properly dense sampling, the transition estimates are wildly inaccurate. Crucially, none of the other phylogenies had adequate sampling within or even across the five focal clades (many only had a handful of representatives from one or two of the focal clades), rendering them unsuitable for our purposes. At the suggestion of the reviewer, we also downloaded a tree from timetree.org for all of Actinopterygii. However, the rampant polytomies in this phylogeny make comparative analyses impossible. We are not aware of any other resource that would allow us to incorporate topological uncertainty. Therefore, we are limited to the published Rabosky et al. phylogeny, which is the most comprehensive time calibrated phylogeny of fishes at this time.

All of that being said, we have made efforts to alleviate some of these concerns because they are indeed valid. Firstly, we had originally removed all families from the analyses that were represented as non-monophyletic in the phylogeny, as we also recognized that this was a potential issue. We now note that the families are monophyletic in the text (line 136). Secondly, we have now implemented a congruification approach to redate the nodes in the Rabosky phylogeny based on two different published phylogenies with fossil calibrations (Alfaro et al. 2018 and Ghezelayagh et al. 2022). This is to account for some of the uncertainty in the divergence times while retaining the dense species sampling necessary for this study. With these redated versions of the Rabosky tree, we reran our analyses and found very consistent results (Figs. S5 and S6). This implies our results are robust to differences in branch lengths, which are more likely to bias rate estimates than topological differences (Purvis et al. 1994). We also reran the non-clade-based Mk analyses with the Ghezelayagh et al. (2022) phylogeny, with expected results considering the lack of polar species in this tree that match our dataset (Fig. S7). Overall, we find consistent results regardless of the dating scheme used, which implies that our findings are relatively robust to phylogenetic uncertainty. Finally, to be as transparent as possible with our study, we now detail in the discussion that the inability to account for phylogenetic uncertainty is an unfortunate limitation of many macroevolutionary studies, including this one.

Ultimately, I believe evolutionary biologists using phylogenetic comparative methods must raise the 'quality bar' by accounting for potential sources of error in comparative inferences, arising not only from tree uncertainty but also from (ecological) data error (e.g., Silvestro et al., 2015). One solution I see is to build or use other trees (e.g., Open Tree of Life or TimeTree of Life) and assess the extent to which phylogenetic error may have affected their comparative inferences, including (critically) estimates of tip-specific speciation rates. It is also key to revisit the inverse latitudinal gradient identified by the previous study, which underpins all of their analyses. Species-level ecological data obtained from large-scale databases (FishBase) or secondary sources (Rabosky et al.) can also be unreliable. This can be validated by using depth and latitude information for a random selection of species in their dataset extracted from the primary literature.

We appreciate this feedback. Please see the response above regarding how we have accounted for phylogenetic uncertainty in the revised version of the manuscript.

We respectfully disagree about unreliability of the latitude data, which come directly from the Rabosky et al. (2018) paper. They generated species geographic range maps using an environmental envelope approach in AquaMaps that predicts species distributions based on available occurrence records and incorporates known environmental predictors. They then consulted taxonomic experts and the primary

literature to evaluate the validity of these ranges. These ranges were then *further* validated by downloading occurrence data from four sites: Global Biodiversity Information Facility (GBIF), Ocean Biogeographic Information System (OBIS), Fishnet2 and VertNet. Occurrence records were cleaned and then used to corroborate the species Aquamaps geographic ranges. These methods combine the most modern, state-of-the-art techniques for identifying species distributions at the macroevolutionary scale using verifiable occurrence data. Therefore, in light of the extremely thorough methods used to validate these data by Rabosky *et al.*, we believe the underlying latitudinal data are reliable for our purposes. Using primary literatures to further validate these distributions would be somewhat circular, as most modern estimates of species ranges at scale also rely on these same databases (OBIS, GBIF, etc.) for the underlying occurrence data, particularly for marine species (Melo-Merino et al. 2020). Those that do not rely on these sources tend to be from studies that are more locally or regionally restricted, which is also not suitable for our global analyses. Finally, we consulted with Lewis Barnett, an expert in fish distributional modelling, who confirmed that these databases are widely used and adequate for these purposes, particularly at a global scale where the potential for diminishing returns of validating individual species ranges (in this case for over 4,000 species) is high. We have expanded our description of the dataset and methods used by Rabosky et al. for obtaining these data in the revised version of our manuscript.

With regards to the depth data downloaded from Fishbase, we note that all the aggregated species-specific depth data come from observations reported in documented sources and thus have biological merit. We did review the data for egregious errors and altered the depths for the two species that clearly had erroneous data, *Lampanyctus jordani* and *Gymnoscopelus hintonoides*, which we have now clarified in the methods of our revised manuscript. Nevertheless, this demonstrates that there are sometimes inaccuracies in the reported depths and binning species into broad depth categories was expected to largely mitigate these issues. In response to this feedback, we did take a random sample of 100 species and validate those depth measurements with other sources and primary literature (Table S2). While there were slight differences in some of the reported depths between Fishbase and other sources, they largely corroborated the Fishbase depths and nearly all species remained in the same depth category. The one exception was *Lyconus brachycolus*, which had a Fishbase maximum depth of 997m, while other sources reported 1,000m. This would recategorize this species from 'intermediate' to 'deep' in our dataset. We were unable to locate a copy of the depth source used by Fishbase to validate the 997m. However, given that our validation procedure largely confirmed the sources that Fishbase uses to obtain depths for other species, we suspect that the sources we have used rounded the exact depth of capture up to 1,000m and that 997 is a more accurate estimate of the known maximum depth. Still, we acknowledge that any approach, including our own, is vulnerable to sampling error, particularly in rare species and those near depth zone boundaries and we now clarify these potential issues in the methods (line 127-129).

Other points:

Global community structure comparisons: "Therefore, in tropical latitudes the shallow fish community strongly differs from the deeper fish community, reflecting limited diversification along the depth axis" & "High-latitude communities appear more phylogenetically homogenous along the depth axis, likely reflecting repeated diversification across depth categories." The PCD metric used for this analysis simply compares phylogenetic similarity/dissimilarity between communities at different depths within each latitude. Any observed dissimilarities will be the result of two factors: transition and diversification rates. Please clarify how the PCD metric can tease these factors apart.

We apologize for the ambiguity of the sentence, as the PCD metric cannot distinguish between these two factors. We have reworded the sentence accordingly: "High-latitude communities appear more phylogenetically homogenous along the depth axis, reflecting repeated invasions of new depth zones."

"We produced 100 simmaps with "ARD" (all rates different) to model transition probabilities between depth states based on an initial log-Likelihood comparison from a sample of reconstructions." Why not first compare the fit of different models and then chose the best fit model(s) to reconstruct depth?

We did initially compare the fit of different models here: "...based on an initial log-Likelihood comparison from a sample of reconstructions". We have reworded the sentence for clarity.

"We then used the fitMk function in phytools 37 to fit Mk models 47,48, a model of discrete character evolution, with equal rates, symmetric rates, and all rates different transition models, and compared the fit of these different transition rate models using AIC." Related to the above point, why is this analysis done for latitude but not depth?

This analysis was done for both latitude and depth. We now clarify this in the revised version of the manuscript.

The information presented in the M&M section under "Latitude and Depth Transitions" is poorly organized. For example, it's unclear why estimates of diversification rates and their relationship with depth go in this section. Without a more structured section layout, it's hard to keep track of all the different analyses and how they are connected.

Upon re-reading the manuscript, we agree that the section was poorly organized and confusing. We have reorganized the structure of the method and results, grouping together all clade-based analyses and non-clade-based analyses under different headings.

Figure 1. "(green: greater than expectation, grey: within expectation, yellow: below expectation)" For ease of interpretation, please embed a symbol box with this info. onto the figure.

Done.

Figure 4 & elsewhere. "High latitude fish communities are more phylogenetically similar throughout the water column, whereas in the tropics, shallow and deep communities differ substantially." While this interpretation is indeed supported by the plot, there are also many more outliers (greater dispersion of data points) in high vs. low latitudes, particularly towards the left (negative latitude values). I would like to see an interpretation for this.

We suspect the drastic difference in variance of PCD values in the Southern Hemisphere has to do with the distribution of landmasses between the Northern and Southern Hemispheres. There is a distinct drop in landmass near -60 degrees latitude and this coincides with the reduced dispersion of PCD estimates, followed by higher PCD estimates nearer to the poles shown in Fig. 4. This mechanism is also likely why the variance of PCD values is higher at high latitudes. As continental shelves are known to coincide with high species diversity, the contrast between continental shelves and the open ocean is more apparent at higher latitudes where there are also reduced barriers to speciation along the depth gradient. We have added this explanation to the results of the revised manuscript (lines 321-325).

Figures S2, S3 & S7 are in my opinion too important to have them buried in the SM. Please consider moving them to the main text (if display items are at the limit, multiple panels can always be combined into a single fig.)

We have combined the previous Fig. S7 with Fig. 3 in the main text. Upon closer inspection we realized that Fig. S2 and S3 were slightly redundant (the BAMM rates are the speciation rates that were originally shown in Fig. S2). We have also expanded Fig. 3 in the main text of the manuscript to include the information previously contained in Figs. S2 and S3.

"We demonstrate that clades that with the highest," Typo.

We have removed the extraneous "that".

"This pattern also extended to several other families, such as lanternfishes (Myctophidae), jack

(Carangidae).” Should be ‘jacks.’
We have made this change.

Reviewer #3 (Remarks to the Author):

This manuscript presents a study that finds evolutionary transitions in oceanic depth contribute to the recently described inverse latitudinal gradient in ray-finned fish speciation rates. This counterintuitive pattern—highest rates of speciation at high latitudes despite highest diversity at low latitudes—defies classic ideas about the tropics as a center for diversification, and the identification of a potential mechanistic explanation is exciting. The authors show that rapidly speciating, high latitude clades exhibit exceptionally high rates of transition between depth zones supporting the predicted association between depth and speciation rate at high latitudes. More generally, they also demonstrate that transitions between depth zones occur more frequently at higher latitudes compared to the tropics and highlight a central role for evolutionary lability of depth in the diversification of fishes. The analyses are thorough, the manuscript is well written, and the figures provide especially clear illustrations of the central results. Despite this promise, however, I have several conceptual and methodological concerns.

We appreciate the positive feedback and thank the reviewer for their time and careful reading of the manuscript. We address the conceptual and methodological concerns in detail below.

1) I would like the authors to more clearly explain the hypothesized dynamics of diversification with depth transitions. It seems that there are two possible mechanisms, and I was unclear on which was thought to be at work. First, shifts in depth might prompt speciation because adaptation to novel environmental conditions (particularly in the deep ocean) leads to reproductive isolation. So, depth transitions would tend to be associated with speciation events, and a but I think more explicit differentiation between mechanism and predicted evolutionary outcomes is required. As I describe in my next comment, character state-associated diversification models could be used to disentangle these different effects and could potentially provide novel insights, such as why some lineages with depth lability exhibit high speciation rates but others do not.

We apologize for the confusion and lack of clarity in the setup and questions of this study. Deep water environments have been shown elsewhere to not correlate with rates of speciation in fishes (Rabosky et al. 2018); therefore, many of the SSE models suggested below are not applicable to our hypotheses. In essence, we predict that speciation rate is associated with rates of depth evolution and now state this hypothesis clearly in the introduction. Rates of depth evolution may or may not also be correlated with rates of trait divergence depending on the exact speciation mechanism, and we emphasize that different mechanisms are not mutually exclusive. We explore this distinction in greater detail in the revised discussion. Determining the exact speciation mechanism along the depth gradient is beyond the scope of this study and is likely not generalizable (it is probably context or lineage dependent). At this scale, we would have no expectation that a universal mechanism or even a single ecological variable (habitat, diet, temperature, etc.) would be correlated with depth evolution in every lineage. We also now state this in the revised version of the manuscript (Lines 355-358). However, it would indeed be an interesting question to pursue in a future study. We have now added much discussion on the topic, with a particular focus on how the biology and ecology of the focal lineages may contribute to rapid depth evolution. We have also substantially revised the introduction to clarify our questions and hypotheses that we address in this study.

2) The authors should consider applying models that simultaneously account for character state-dependent diversification in their estimation of depth transition rates. Estimation of character state transitions that take the phylogeny as given can be biased when that character influences diversification (Maddison 2006). For this reason, I am somewhat concerned about the robustness of the result that

depth transition rates tend to be higher in rapidly speciating and/or high latitude clades. Depending on the hypothesized dynamics of depth and speciation (see previous comment), it may be more appropriate and insightful to use a model like BiSSE (Maddison et al. 2007), MuSSE (FitzJohn 2012), or HiSSE (Beaulieu and O'Meara 2016) if depth spurs speciation by providing ecological opportunity, or BiSSE-ness (Magnuson-Ford et al. 2012) if depth transitions are associated with speciation events.

Here, we hypothesize that speciation events are associated with changes in depth, rather than a specific depth zone (i.e., deep habitats) driving high rates of speciation. We apologize for the lack of clarity in our hypotheses and have significantly revised the manuscript to avoid future confusion. Given this hypothesis, the only applicable methods would be BiSSE-ness or ClaSSE. We understand that the reviewer would like a more direct analysis associating speciation events and depth transitions and, theoretically, such an analysis would be useful for these purposes. However, there are a few known issues with SSE models more generally, which complicates these analyses. Firstly, BiSSE-ness requires a binary state. This is fine enough, as we can (and did) combine the two deeper depth categories, making two total depth categories, though this does not grant us the preferred resolution. Secondly, and perhaps most concerning, studies have come out detailing the potential for inflated Type 1 error in SSE models (Rabosky and Goldberg 2015; Beaulieu and O'Meara 2016). This is because traditional SSE models do not account for background rate variation; thus, they assume that all rate heterogeneity is attributed only to differences between character states. Therefore, shifts in speciation rates in a single lineage that also has a character shift can misattribute the speciation shift to the character more generally across the tree. Combined with the fact that SSE models compare against unrealistic null models (Rabosky and Goldberg 2015; Beaulieu and O'Meara 2016; Caetano et al. 2018), this means that such analyses, particularly across a large tree with complex variation in states and rates, will virtually always favor the more complex model. Although recent advances have been made to accommodate hidden states in SSE models (e.g., Caetano et al. 2018), which have alleviated some concerns, these model extensions have not yet been applied to BiSSE-ness or ClaSSE. Furthermore, we have conferred with an expert in SSE models, Daniel Caetano, who has confirmed the issues mentioned above for BiSSE-ness models and the lack of published extensions for these models to mitigate Type-1 error.

All of that being said, we are not aware of any alternative published methods for these types of analyses that ameliorate the concerns detailed above. Therefore, we did run the BiSSE-ness model. As expected, we find that the full cladogenic model is unequivocally favored over an anagenic model using a maximum-likelihood optimization approach in *diversitree* (dAIC = 223), indicating that depth changes are driving cladogenic events. These results do support our interpretations and the overall story would remain unchanged with this additional analysis. Nevertheless, we emphasize that it is not clear that any alternative results with a macroevolutionary dataset as complex as this one are possible. Thus, we are hesitant to place too much emphasis on this analysis and do not include it in the revised version of the manuscript.

We believe this comment is partially motivated by the fact that the previous version of the manuscript assumed the phylogeny as given. We have now accounted for phylogenetic uncertainty in the revised version of the manuscript and find very similar results, implying this study is robust particularly to variation in divergence time estimates, which would be most likely to influence our results. Please see our response to the reviewers above and lines 164-172 in the manuscript.

3) Related to the two comments above, I am also somewhat concerned by the use of a clade-based approach for investigating diversification dynamics. The sampling units for the first part of the analysis are named clades, but this partitioning of the phylogeny is somewhat arbitrary. This approach may obscure the role of depth as other unconsidered factors that are shared within a clade can also contribute to diversification. While the association between depth transitions and speciation rate is repeated across multiple high latitude clades, there are a couple of results that complicate inferences about a causal role for depth. First, the clades that drive the relationship are relatively closely related; when phylogeny is accounted for in the regression of speciation and depth transition rates, the relationship is no longer significant (lines 222-224). To be fair, the authors are aware of this complication and address it to some

extent in the Discussion. The second complicating result is that several additional clades with high rates of depth transitions do not show high speciation rates. Combined, these results suggest that the effect of depth is perhaps not pervasive but rather context or lineage dependent. I recommend the use of state-based diversification methods (see comment 2) may clarify the generality of the depth's effect on diversification across lineages.

We agree with the reviewer that the effect of depth can be context/lineage dependent and detail this in the manuscript (lines 355-358, 410-415). Furthermore, our robust analyses include various methods that account for phylogenetic uncertainty, hidden states, and non-clade-based estimates of the relationship between depth lability and latitude. Though we do not include the additional BiSSE-ness analysis in the revised manuscript for the reasons stated above, these analyses further support our findings. The suite of SSE models currently available do not allow us tease apart clade-specific trends, which would be most relevant to our first hypothesis. Our analyses that do not use clades as a unit of sampling are remarkably consistent with the clade-based trends, implying our results are robust to variation in partitioning scheme. In the revised version of the manuscript, we have added further emphasis that these results are likely context- and lineage-dependent. This does not detract from the overall interpretations of the study, and, in fact, serves as an interesting catalyst for future studies to investigate the unique properties that allow perciform lineages to dominate both high latitudes and the depth gradient (lines 436-440). Our findings emphasize that there is something special about the combination of the five focal lineages, rates of depth evolution, and high-latitudes, likely facilitated by the reduced abiotic gradient at high latitudes. While we recognize that clades are somewhat arbitrary units of partitioning a phylogeny, the remarkable signal in Perciformes is a testament to the success of this order and represents yet another primary discovery of this study. We hope that this study encourages research into the evolution of perciform fishes.

4) I recommend further evaluation of the robustness of these results to phylogenetic uncertainty. While I applaud the authors consideration of alternative categorizations of depth and analytical approaches, the phylogeny is treated as known. However, error in estimation of divergence times and poor resolution in some nodes may have considerable influence on estimates of evolutionary rates (of depth and latitude transitions as well as speciation) and therefore influence inferred relationships between depth and latitude transitions and speciation. Repeating analyses on a sample of plausible phylogenetic trees could help address this issue, though I recognize some of the methods used in this study are computationally demanding and this strategy may be time intensive. Nevertheless, some assessment of the role of phylogenetic uncertainty is necessary to demonstrate that key results are generalizable beyond this phylogenetic estimate.

We acknowledge this limitation of our study and have made efforts to alleviate some of these concerns in the revised version of this manuscript with the published resources available. Please see our response above to reviewer 2.

5) A final, minor comment is that I would like the authors to justify the use of different ways of representing species depth in different analyses. Estimates of rates of depth transitions are based on species deepest occurrence (lines 101-103) but tests of association between latitude and rates of depth evolution involve species' median depths (lines 161-163). Also, do these alternative descriptions of depth alter interpretation at all?

This is a good point. We had originally used maximum depth to categorize species because many deep-sea fishes span both shallow and deep zones of the ocean and the shallow portion of their depth range is more easily discovered, giving an incomplete view of an extinct species' maximum depth. Nevertheless, we have repeated our analyses with species re-categorized based on their median depths and our findings remain consistent (Fig. S4).

Citations

- Alfaro, M. E., B. C. Faircloth, R. C. Harrington, L. Sorenson, M. Friedman, C. E. Thacker, C. H. Oliveros, D. Černý, and T. J. Near. 2018. Explosive diversification of marine fishes at the Cretaceous–Palaeogene boundary. *Nat Ecol Evol* 2:688–696.
- Beaulieu, J. M., and B. C. O'Meara. 2016. Detecting Hidden Diversification Shifts in Models of Trait-Dependent Speciation and Extinction. *Systematic Biology* 65:583–601.
- Betancur-R., R., R. E. Broughton, E. O. Wiley, K. Carpenter, J. A. López, C. Li, N. I. Holcroft, D. Arcila, M. Sanciangco, J. C. Cureton II, F. Zhang, T. Buser, M. A. Campbell, J. A. Ballesteros, A. Roa-Varon, S. Willis, W. C. Borden, T. Rowley, P. C. Reneau, D. J. Hough, G. Lu, T. Grande, G. Arratia, and G. Ortí. 2013. The Tree of Life and a New Classification of Bony Fishes. *PLoS Curr*, doi: 10.1371/currents.tol.53ba26640df0ccae75bb165c8c26288.
- Caetano, D. S., B. C. O'Meara, and J. M. Beaulieu. 2018. Hidden state models improve state-dependent diversification approaches, including biogeographical models: HMM AND THE ADEQUACY OF SSE MODELS. *Evolution* 72:2308–2324.
- Ghezelayagh, A., R. C. Harrington, E. D. Burress, M. A. Campbell, J. C. Buckner, P. Chakrabarty, J. R. Glass, W. T. McCraney, P. J. Unmack, C. E. Thacker, M. E. Alfaro, S. T. Friedman, W. B. Ludt, P. F. Cowman, M. Friedman, S. A. Price, A. Dornburg, B. C. Faircloth, P. C. Wainwright, and T. J. Near. 2022. Prolonged morphological expansion of spiny-rayed fishes following the end-Cretaceous. *Nat Ecol Evol* 1–10. Nature Publishing Group.
- Hughes, L. C., G. Ortí, Y. Huang, Y. Sun, C. C. Baldwin, A. W. Thompson, D. Arcila, R. Betancur-R., C. Li, L. Becker, N. Bellora, X. Zhao, X. Li, M. Wang, C. Fang, B. Xie, Z. Zhou, H. Huang, S. Chen, B. Venkatesh, and Q. Shi. 2018. Comprehensive phylogeny of ray-finned fishes (Actinopterygii) based on transcriptomic and genomic data. *Proc Natl Acad Sci USA* 115:6249–6254.
- Melo-Merino, S. M., H. Reyes-Bonilla, and A. Lira-Noriega. 2020. Ecological niche models and species distribution models in marine environments: A literature review and spatial analysis of evidence. *Ecological Modelling* 415:108837.
- Paulus, E. 2021. Shedding Light on Deep-Sea Biodiversity—A Highly Vulnerable Habitat in the Face of Anthropogenic Change. *Frontiers in Marine Science* 8.
- Purvis, A., J. L. Gittleman, and H.-K. Luh. 1994. Truth or Consequences: Effects of Phylogenetic Accuracy on Two Comparative Methods. *Journal of Theoretical Biology* 167:293–300.
- Rabosky, D. L., J. Chang, P. O. Title, P. F. Cowman, L. Sallan, M. Friedman, K. Kaschner, C. Garilao, T. J. Near, M. Coll, and M. E. Alfaro. 2018. An inverse latitudinal gradient in speciation rate for marine fishes. *Nature* 559:392–395.
- Rabosky, D. L., and E. E. Goldberg. 2015. Model Inadequacy and Mistaken Inferences of Trait-Dependent Speciation. *Systematic Biology* 64:340–355.

REVIEWER COMMENTS

Reviewer #2 (Remarks to the Author):

While I commend the authors for the through revision of their paper, I'm still skeptical that the phylogenetic comparative results of the study are reliable as they are based on an extremely poor estimate of the phylogeny of ray-finned fishes. Recalibrating the tree, while important, does not overcome its main issues. I gave a summary of the topological issues of this tree at the family level, but this is far from being the most critical problem. Again, many 'good genera' are found to be polyphyletic in this megatree. Since trees from timetree.org or opentreeoflife.org are not viable options, my suggestion is to use a couple of reliable backbone trees and increase their sampling using sequence data from GenBank. This can be done by implementing backbone constraint ML searches. First, download some of the most commonly used nDNA and mtDNA legacy markers to cover the ~4k species sampling. Then, run the backbone constraint analyses (need sequences for all species in the backbone tree; if some are missing, then prune them out of the backbone before running the ML analyses). Spend some time curating the resulting trees and remove rogue taxa (e.g., miss IDs, contaminated sequences). Then use again congruification to calibrate the resulting phylograms. With the new trees, test not only the latitudinal-depth diversification gradient hypothesis (this study), but also the inverse latitudinal gradient hypothesis of Rabosky et al., which again underpins all major results. While this may seem time consuming, luckily ~4k species is just a small fraction of the ray-finned fish tree, and in my own experience it can be done. I must insist that the quality bar for macroevolutionary inferences must be raised by accounting for many sources of error in comparative inferences, in particular tree uncertainty.

Reviewer #3 (Remarks to the Author):

I appreciate the authors' attention to the concerns I raised in my previous review. This version of the manuscript has improved substantially in several ways. First, the hypothesized link between depth and speciation rates is now clearer, and the focus on depth transitions as a driver (rather than ecological opportunity provided by particular depth zones) alleviates some of my methodological concerns (i.e., because the hypothesis does not posit diversification by ecological opportunity within depth zones, the state-dependent diversification models are not necessary). Second, I appreciate the detailed examination of the nature of depth transitions within some of the most rapidly diversifying clades. These biological details nicely enrich the story and provide context to this broad scale analysis. Third, the inclusion of alternative phylogenetic trees in the analysis shows that these patterns are (to some extent) robust to phylogenetic uncertainty.

I have only a couple of additional comments.

1) Although I accept the authors' justification for their clade-based approach over the BiSSE-ness model for evaluating the relationship between depth transitions and speciation rates, I recommend including a statement about their rationale in the main manuscript. The explanation in the rebuttal letter is reasonable and well justified, but the main manuscript contains no mention of the alternative method or the explanation for the authors' methodological preference. I imagine other readers will have concerns similar to those I raised in my earlier review, and for that reason, I think it would be useful for the authors to be explicit about the basis for their choice.

2) Overall, the manuscript is now much clearer about depth transitions leading directly to speciation rather than depth transitions providing ecological opportunity and facilitating diversification within depth zones. There is nevertheless one line in the text that seems to confound these processes: in the Discussion on lines 363-364, the authors describe depth zones as providing "a wide array of habitats and associated ecologies, providing ample opportunity for diversification." This sentence seems to set up a review of available niches and diversification within depth zones, and caused me a bit of

confusion at first. The text that follows, however, provides discussion (and a thoughtful one) of how depth transitions occur within focal clades. So, as a whole, the paragraph fits well within the context of depth transitions driving speciation, but the sentence I mention above should probably be reworded to more accurately set it up.

REVIEWER COMMENTS

Reviewer #2 (Remarks to the Author):

While I commend the authors for the through revision of their paper, I'm still skeptical that the phylogenetic comparative results of the study are reliable as they are based on an extremely poor estimate of the phylogeny of ray-finned fishes. Recalibrating the tree, while important, does not overcome its main issues. I gave a summary of the topological issues of this tree at the family level, but this is far from being the most critical problem. Again, many 'good genera' are found to be polyphyletic in this megatree. Since trees from timetree.org or opentreeoflife.org are not viable options, my suggestion is to use a couple of reliable backbone trees and increase their sampling using sequence data from GenBank. This can be done by implementing backbone constraint ML searches. First, download some of the most commonly used nDNA and mtDNA legacy markers to cover the ~4k species sampling. Then, run the backbone constraint analyses (need sequences for all species in the backbone tree; if some are missing, then prune them out of the backbone before running the ML analyses). Spend some time curating the resulting trees and remove rogue taxa (e.g., miss IDs, contaminated sequences). Then use again congruence to calibrate the resulting phylograms. With the new trees, test not only the latitudinal-depth diversification gradient hypothesis (this study), but also the inverse latitudinal gradient hypothesis of Rabosky et al., which again underpins all major results. While this may seem time consuming, luckily ~4k species is just a small fraction of the ray-finned fish tree, and in my own experience it can be done. I must insist that the quality bar for macroevolutionary inferences must be raised by accounting for many sources of error in comparative inferences, in particular tree uncertainty.

We greatly appreciate the time and expertise in consideration of our revised manuscript. However, it is not clear what specific topological issues the reviewer is referring to with the Rabosky et al. (2018) phylogeny, and we can find no such record of these claims after a thorough inspection of the phylogeny and associated literature. We have also consulted Dr. Thomas Near, one of the original authors of the phylogeny and an expert in constructing supertrees for fishes, who is unaware of such critiques. As we describe below, we believe requiring a new phylogeny is both unreasonable and unwarranted and hope that the request will be reconsidered in light of the additional details provided and new sensitivity analyses we have conducted.

- 1) **Reconstructing a phylogeny by querying sequences from GenBank would be tautological.** The phylogeny of Rabosky et al. (2018) is already constructed from sequence data across 24 genes mined directly from GenBank and other sources (e.g., Euteleost Tree of Life project, Barcode of Life). Any sequences that were added to their dataset have since been uploaded to the NCBI GenBank database upon publication. Therefore, querying GenBank for sequences to generate a new phylogeny would simply be recreating the efforts of Rabosky et al., likely with decreased resolution as we would only be focusing our efforts on a single database and do not have access to their custom bioinformatics pipelines and computational resources. Any additional sequences on GenBank that have been added since publication are unlikely to substantially add to our dataset, particularly when the focus of this study is on clades for which it is generally difficult to obtain tissue samples (e.g., Zoarcidae, Liparidae, Notothenioids). This is very much apparent in our difficulties to find another already-published phylogeny suited to our purposes, as there are no other currently published trees with adequate species sampling across the five focal clades needed for this study. In short, this would constitute a circular task to recapitulate a previously published study.
- 2) **The amount of work requested would constitute an entirely new publication.** Building supertrees is a very complex, nuanced, and time-consuming process if done properly. There are countless non-trivial steps and philosophical questions that one must grapple with, ranging from the sequence alignment and partitioning scheme to the proper substitution models and different tree reconstruction models (Garamszegi 2016). To properly select models, one must take an iterative approach, modeling the data under the available possibilities to select the best fitting model, adding to the computational time and resources necessary. Furthermore, multiple error points and biases, such as long-branch attraction, saturation, and gene tree discordance also need to be carefully considered after tree construction. Every one of these steps is considerably more complicated on supertrees with widespread rate heterogeneity spanning millennia, as would necessary for macroevolutionary studies such as this one. In their recent review of the topic, Zaharias and Warnow (2022) state, "the past decade has produced methods for alignment and phylogeny estimation that have excellent accuracy on small- to moderate-sized datasets, but only a few of these methods can analyse even moderately large datasets (1000 sequences)". Our dataset would necessitate analysis of multiple sequences from over 4,000 species. Lastly, these steps ignore the initial – and very essential – data filtering and quality-checking steps necessary when using data from GenBank, which is notoriously messy. All told, the time required to properly construct a new phylogeny for thousands of species would be prohibitive and require considerable computational resources. Moreover, reviewer 2 has requested that we

re-run the original analyses of Rabosky et al. (2018) on this newly developed tree. We note that we do not have access to their scripts to rerun the core analyses and would have to regenerate much of the code from scratch, effectively duplicating the entire workload of Rabosky et al. in addition to our current study.

- 3) **We have already addressed and acknowledged the unavoidable concerns of topological uncertainty, and in this revision provide additional sensitivity analyses documenting the robustness of our results to topological variation.** We note that in our prior revised submission we did already run one of our analyses on a different phylogeny (Ghezelayagh et al. 2022), which yielded very consistent results to those run on the Rabosky et al. phylogeny. This result already underscores the robustness of our findings and the general patterns of our study to both topological and branch length uncertainty. In response to reviewer #2's concerns, we have now implemented a sensitivity analysis to demonstrate the robustness of our clade-specific results to topological variation. We find that our results are remarkably consistent, even with just 40% of species randomly selected from the phylogeny (see lines 172-177 in the revised manuscript; Fig. S7). Lastly, we have fully and openly acknowledged the unavoidable issues of accounting for phylogenetic uncertainty for our clade-specific analyses both in our responses to the reviewers and in the text of the manuscript. This is a hurdle that plagues modern macroevolutionary research and though it is laudable to hold the field to higher standards (and we fully support those efforts), it is not always possible to achieve those goals with the resources available. We do not believe that this makes the research any less important or impactful. And, more specific to this study, we affirm that re-doing the same analysis as Rabosky et al. (2018) will not alter our results.

Reviewer #3 (Remarks to the Author):

I appreciate the authors' attention to the concerns I raised in my previous review. This version of the manuscript has improved substantially in several ways. First, the hypothesized link between depth and speciation rates is now clearer, and the focus on depth transitions as a driver (rather than ecological opportunity provided by particular depth zones) alleviates some of my methodological concerns (i.e., because the hypothesis does not posit diversification by ecological opportunity within depth zones, the state-dependent diversification models are not necessary). Second, I appreciate the detailed examination of the nature of depth transitions within some of the most rapidly diversifying clades. These biological details nicely enrich the story and provide context to this broad scale analysis. Third, the inclusion of alternative phylogenetic trees in the analysis shows that these patterns are (to some extent) robust to phylogenetic uncertainty.

Thank you for the helpful feedback. We appreciate that the reviewer finds the revised manuscript much improved.

I have only a couple of additional comments.

1) Although I accept the authors' justification for their clade-based approach over the BiSSE-ness model for evaluating the relationship between depth transitions and speciation rates, I recommend including a statement about their rationale in the main manuscript. The explanation in the rebuttal letter is reasonable and well justified, but the main manuscript contains no mention of the alternative method or the explanation for the authors' methodological preference. I imagine other readers will have concerns similar to those I raised in my earlier review, and for that reason, I think it would be useful for the authors to be explicit about the basis for their choice.

This is a good point and we have added a brief justification regarding our methodological preferences in the text.

Lines 178-181: Though there are existing methods to more directly assess the association between depth transitions and cladogenetic events, there are methodological concerns, particularly for large-scale studies with substantial rate heterogeneity, which make these analyses ill-suited for our purposes.

2) Overall, the manuscript is now much clearer about depth transitions leading directly to speciation rather than depth transitions providing ecological opportunity and facilitating diversification within depth zones. There is nevertheless one line in the text that seems to confound these processes: in the Discussion on lines 363-364, the authors describe depth zones as providing "a wide array of habitats and associated ecologies, providing ample opportunity for diversification." This sentence seems to set up a review of available niches and diversification within depth zones, and caused me a bit of confusion at first. The text that follows, however, provides discussion (and a thoughtful one) of how depth transitions occur within focal clades. So, as a whole, the paragraph fits well within the context of depth transitions driving speciation, but the sentence I mention above should probably be reworded to more accurately set it up.

We agree and have removed the sentence.

Citations

- Garamszegi, L. Z. 2016. *Modern Phylogenetic Comparative Methods and Their Application in Evolutionary Biology: Concepts and Practice*. Springer Berlin Heidelberg.
- Ghezelayagh, A., R. C. Harrington, E. D. Burress, M. A. Campbell, J. C. Buckner, P. Chakrabarty, J. R. Glass, W. T. McCraney, P. J. Unmack, C. E. Thacker, M. E. Alfaro, S. T. Friedman, W. B. Ludt, P. F. Cowman, M. Friedman, S. A. Price, A. Dornburg, B. C. Faircloth, P. C. Wainwright, and T. J. Near. 2022. Prolonged morphological expansion of spiny-rayed fishes following the end-Cretaceous. *Nat. Ecol. Evol.* 1–10. Nature Publishing Group.
- Rabosky, D. L., J. Chang, P. O. Title, P. F. Cowman, L. Sallan, M. Friedman, K. Kaschner, C. Garilao, T. J. Near, M. Coll, and M. E. Alfaro. 2018. An inverse latitudinal gradient in speciation rate for marine fishes. *Nature* 559:392–395.
- Zaharias, P., and T. Warnow. 2022. Recent progress on methods for estimating and updating large phylogenies. *Philos. Trans. R. Soc. B Biol. Sci.* 377:20210244. Royal Society.